# Landowner perceptions of woody plants and prescribed fire in the Southern Plains, USA

**Dianne A. Stroman**[1]*, **Urs P. Kreuter**[2], **Carissa L. Wonkka**[3]

**1** Department of Environmental Science, Collin College, Frisco, Texas, United States of America,
**2** Department of Ecology and Conservation Biology, Texas A&M University, College Station, Texas, United States of America, **3** USDA Agricultural Research Service, Northern Plains Agricultural Laboratory, Sidney, Montana, United States of America

☯ These authors contributed equally to this work.
* dstroman@collin.edu

**Data Availability Statement:** All relevant data are within the manuscript and its Supporting Information files.

**Funding:** UK received finding from the National Science Foundation Dynamics of Coupled Human

## Abstract

Grassland environments face a number of threats including land use change, changing climate and encroachment of woody plants. In the Southern Plains of the United States, woody plant encroachment threatens traditional agricultural grazing economies in addition to grassland dependent wildlife species. Numerous studies have examined the physical drivers of conversion from grassland to woodland but social drivers may be equally important to understanding the causes of and prescriptions for environmental degradation. In this paper, we report the results of a survey of landowners in the Southern Plains of Texas and Oklahoma in which we asked participants to estimate the current amount of woody plant cover on their land, their preferred amount of woody plant cover and about their perspectives regarding the use of prescribed fire for managing woody plants. Prescribed fire is ecologically and economically one of the most effective tools for maintaining grasslands but many landowners do not use this tool due to lack of knowledge, lack of resources and concerns over safety and legal liability. We found that while most of our respondents did express a desire for less woody plant cover on their land, woody plant preference did not affect landowner's use of prescribed fire. However, belonging to a prescribed burn association and owning larger properties were correlated with increased use of prescribed fire. Woody plant cover preference was significantly influenced by landownership motivations, with hunters and other recreational motivated landowners preferring more trees and ranchers preferring fewer. This is important because throughout most of our study area, there has been a steady shift from agricultural production to amenity or recreational landownership, a trend that may undermine efforts to restore or maintain open grasslands. Future outreach efforts to promote prescribed fire to maintain grasslands should more actively support prescribed burn associations, which is an effective vehicle for increasing prescribed fire use by private landowners.

and Natural Systems Program (Contact # DEB-1413900). The sponsors did not play a role in the study design, data collection and analysis, decision to publish or preparation of this manuscript. the funder URL can be found at: https://www.nsf.gov/pubs/2018/nsf18503/nsf18503.htm.

**Competing interests:** The authors have declared that no competing interests exist.

## Introduction

Throughout many historically grassland-dominant ecoregions, there has been a shift towards increasing concentration of woody plant species. Within our study area, the Southern Plains of the USA, increases in woody cover have ranged from 10% in the 1930's to over 30% today, a trend that is particularly prevalent in areas with more development [1]. This conversion from grassland to woodlands often has cascading effects on local floral and faunal communities and can alter abiotic factors such as soils and local hydrology [2,3]. Additionally, grasslands that have converted to woodlands have lower livestock production potential and are more susceptible to wildfires [4]. The reasons behind this vegetation shift include climate change, elimination of fire from the landscape, overgrazing by livestock and shifts in land use [1,5].

The increase in woody plant cover has encouraged the development of a variety of programs at the local, state and federal levels designed to restore grasslands. Examples include the Brush Busters program, Prescribed Burn Associations (PBAs), Texas Parks and Wildlife Department's Habitat for Upland Birds, and the Natural Resources Conservation Service EQIP program [6,7]. These programs provide financial support, training, education and other incentives to combat woody plant invasion on grasslands. While such programs play an important role for landowners who need help maintaining productive grasslands, the overall problem of woody plant expansion in grasslands continues to increase.

In this paper, we report the results of a survey of landowners in the Southern Plains regions of Texas and Oklahoma in which we asked the survey participants to estimate the amount of woody plant cover on their rural land, and about their perspectives regarding the use of prescribed fire as a woody plant management tool. Accurately estimating woody cover density is difficult from ground level because visual foreshortening of dispersed trees seen from that perspective may suggest higher woody plant density than overhead measurements of canopy cover would indicate. Overestimation of the scope of a problem may lead some people to assume it is too difficult or too risky to deal with, leading to inaction [8]. On the other hand, underestimation of a problem can also lead to inaction because the issue does not seem urgent. We also asked landowners what the *preferred* amount of woody cover is for their land. Knowing landowners' preference for woody plant cover is important because if landowners prefer more woody plants, then they are unlikely to reduce them or seek outside assistance designed to restore grasslands.

We also examine whether woody plant cover preferences influenced landowner use of prescribed fire. While prescribed fire has been found to be ecologically and economically one of the most effective tools for maintaining grasslands [4,9,10], many landowners do not use this land management tool due to lack of knowledge and resources to apply it safely and concerns over legal liability [11,12]. Here, we explore if landowners' preferences for woody plant cover may be an additional social driver of landscape scale transition from grasslands to woodlands.

We address three key questions. 1) Do landowner perceptions about woody plant expansion correspond with estimated changes in their area? 2) Do landowners within the Southern Plains desire less woody cover on their land? 3) Do expressed woody cover preferences influence the adoption of prescribed fire on private land?

## Methodology

### Study area

The study reported here was conducted in the Southern Plains region of Texas and Oklahoma (Fig 1), where the dominant invasive woody plants are native mesquite (*Prosopis glandulosa*), Ashe juniper (*Juniperus ashei*), redberry juniper (*J. pinchotii*), eastern redcedar (*J. virginiana*)

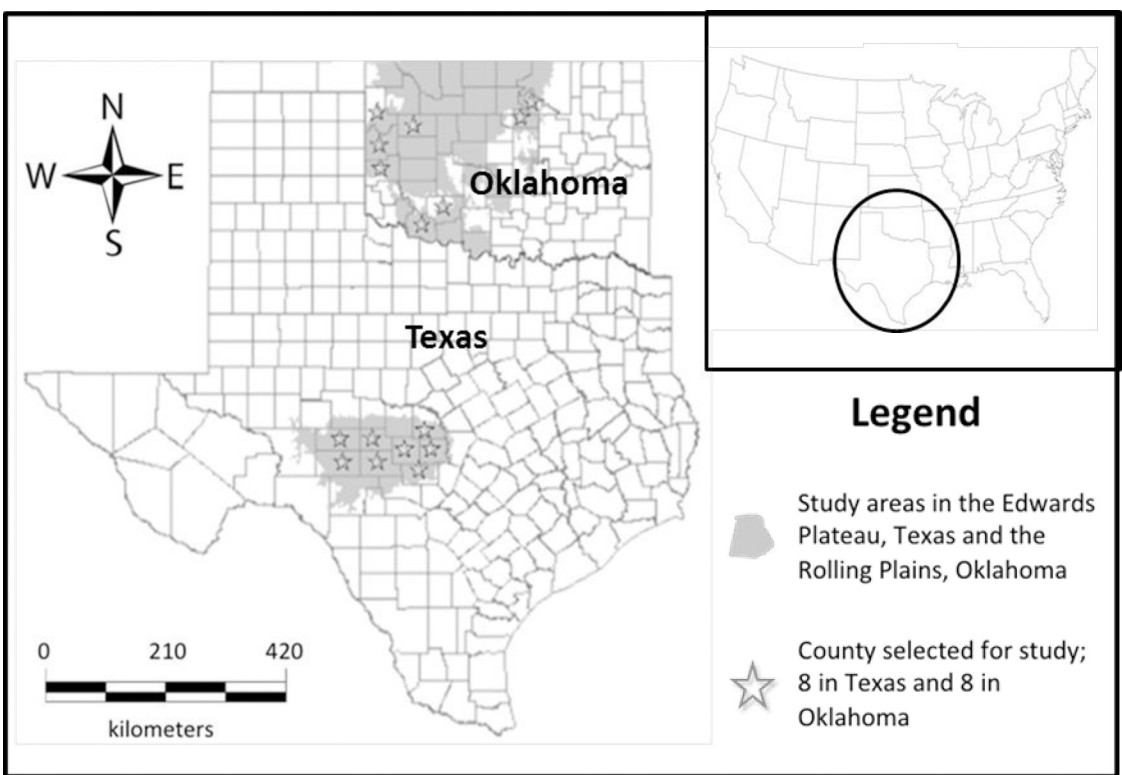

**Fig 1. Map of study area.** Ecoregions in Oklahoma and Texas are shaded and sample counties in each study area location are indicated with *.

and exotic Chinese tallow (*Triadica sebiferium*). The 16 counties we selected for the study were chosen because they incorporated an active PBA and because the role of prescribed fire was central to the overall study. This allowed us to include both general landowners and PBA members in our survey sample. In Texas, we focused on an eight county region within the Edwards Plateau eco-region in the central part of the state. Those eight counties were: San Saba, Llano, Mason, Gillespie, Kimble, Menard, Sutton and Schleicher. In Oklahoma, our sampled counties were clustered in the eastern and northcentral parts of the Rolling Plains eco-region within the state. The eight counties in Oklahoma included: Beckham, Comanche, Dewey, Ellis, Roger Mills, Tillman, Pawnee and Payne.

## Mail survey

This study included human study participants. It was approved by the Texas A&M University IRB board. Consent came in the form of participants choosing to fill out and return a survey questionnaire. All participant data was analyzed anonymously. The survey included 1,918 landowners in Texas and Oklahoma who owned a minimum of 40.5 ha (100 acres). We applied this minimum size requirement with the assumption that landowners with smaller sized properties are less likely to apply prescribed fire, regardless of their perspectives of woody plant density. This survey sample was derived using a stratified sample selection approach. The first stratum consisted of general landowners included in the tax databases of each of the 16 selected counties. From each of these databases, 100 landowners who owned at least 40.5 ha were randomly selected for a first stratum sub-sample of 1,600 landowners. The second stratum consisted all members of PBA in the 16 selected counties. The population of

PBA members in each of the two states was small enough that our PBA sample consisted the whole subgroup. This resulted second stratum sub-sample of 318 PBA members (126 in Texas and 192 in Oklahoma). To ensure that the PBA members were not a biased subset of the combined sample of 1,918 landowners, we compared responses from the two subsamples regarding perceptions of woody plants and prescribed fire and about the proportion of income derived from the property.

In order to assess landowners' perceptions and preferences concerning woody plant cover and the use of prescribed fire on their land, we asked the survey participants a series of questions as part of a broader mail survey about woody plant expansion and the role of prescribed fire in the Southern Plains (S1 File). During initial development, the survey questionnaire was provided to selected stakeholders for feedback and content recommendations, after which it was revised before mailing.

The survey was administered using a five-phase mail protocol [13] consisting of: initial letter to inform landowners about the study (day 1); survey questionnaire with postage paid return envelope (day 7); reminder/thank-you postcard (day 21); replacement questionnaire to non-respondents (day 42); and a final reminder/thank you postcard to non-respondents (day 56). The survey was conducted in October and November 2015 and completed questionnaires were accepted for 150 days from the survey start date.

A non-response bias analysis was not conducted because alternative contact information (e.g., telephone number) required for such an analysis [13] was not obtainable. In the absence of non-response data, we compared data from early and later respondents (i.e., first and second wave responders) as a means of inferring non-response bias from differences between these two groups [14]. Respondents were included in one of two groups based on receipt of their questionnaire during or after the first 3 weeks from the mailing date of the questionnaire (i.e., first and second wave responders). We used Mann-Whitney rank tests to compare the variable values of these two groups for five key variables: age of respondent; reliance on income from their property; perception of and preference for woody plant cover on their land, and whether or not they thought woody plants were a problem on their land.

## Measuring woody cover perceptions and preferences

Survey participants' perceptions of the amount of woody plant cover on their property was assessed by asking if specific types of woody plants (cedar/juniper–*Juniperus* spp. and mesquite–*Prosopis glandulosa*) had increased on their property in the last ten years. Using a categorical visual question prompt (Fig 2), participants were also asked to estimate the current proportion of trees to grassland and their preferred amount of woody cover for their land.

Using landowners' responses to whether or not woody plant cover had increased on their land, we compared average response rates on a county by county basis to published estimates of changes in woody cover between 2004–2014 [15]. While this did not allow us to assess whether woody cover on individual properties had actually increased during that 10-year period, it did allow us to determine if landowners' perceptions of changes in woody plant cover in a specific county corresponded with cover change data for that county.

To gauge the difference between individual landowner's current cover perceptions and their stated woody cover preferences, we subtracted the numeric value for their answers to the two questions referred to in the visual prompt in Fig 2. By subtracting the respondent's current amount of woody cover estimate from their cover preference a number between -3 and 3 was obtained. Negative numbers indicate respondents' desire for less woody plant cover then they currently have, zero indicates respondents' satisfaction with their current woody plant cover, and positive values indicate that respondents prefer more woody plant cover.

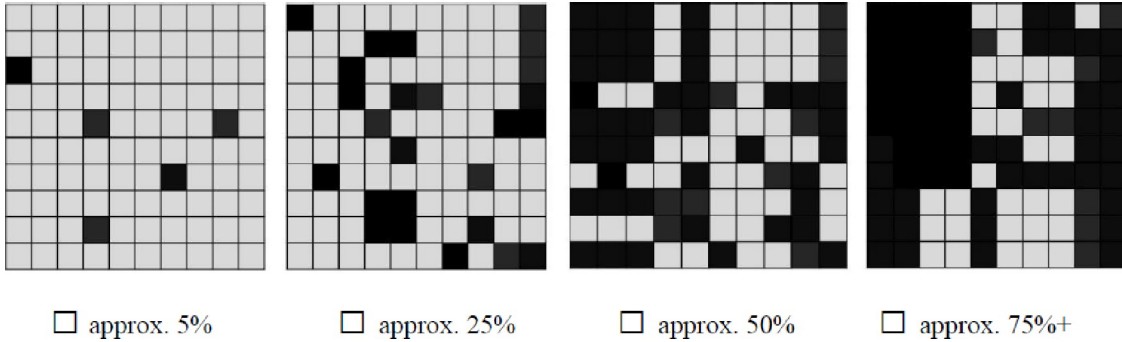

☐ approx. 5%          ☐ approx. 25%          ☐ approx. 50%          ☐ approx. 75%+

**Fig 2. Survey questions used to gauge survey participants' perception of and preference for woody plant cover on their property.** Question prompts were: 1) If you were to look down on your land from above, which picture do you feel would best represent the proportion of trees/shrubs (black boxes) to grassland (gray boxes) found on your property? 2) Which proportion of trees/shrubs (black boxes) to grassland (gray boxes) would you prefer on your property?

## Data and regression analyses

Data were initially analyzed using frequency distribution metrics. We used null values for unanswered questions, resulting in some variation of n values for each variable. We based this on the assumption that data were missing completely at random (MCAR) as we could not determine respondent's reasons for not answering any particular question. Based on this assumption, we did not impute MCAR data.

In addition to the visual question prompt asking respondents about their preferred amount of woody cover (Fig 2), we predicted that a variety of variables would influence woody cover preferences. Using both demographic and landowner specific variables (Table 1) we used ordinal logistic regression models to estimate woody cover preferences. Woody cover preference was used as the dependent variable in our model. Independent variables in our models included demographic controls (age, gender, state of residence) and variables we predicted would influence woody cover preference (residency on property–full time, occasional, absentee; property size; annual income from property; PBA membership; and landownership motivations).

Table 2 reports on the results of the principal components analysis (PCA) used to reduce respondents' landownership motivations into five latent variables for inclusion in the regression model. These latent variables represent the ownership motivation categories listed in Table 1. We used Cronbach's alpha (α) > 0.6 to identify internal reliability of each latent variable [16,17]. These latent variables were used as independent variables in subsequent regression models examining woody plant cover preferences and use of prescribed fire. All statistical analyses were completed using STATA 14 software [18].

In order to test our prediction that landowners who prefer less woody cover on their property will be more likely to use prescribed fire, we developed a regression model using previous use of prescribed fire as the dependent variable. Independent variables for this model included: demographic controls (age, gender, state of residence) and other variables we predicted would influence use of prescribed fire (woody cover preferences; residency on property–full time, occasional, absentee; property size; PBA membership; landownership motivations).

## Results

### Response rates and non-response bias assessment

Of the 1,918 survey questionnaires mailed, 65 were returned as undeliverable, lowering our sample size to 1,853. A total of 680 useable questionnaires were returned representing an

**Table 1. Dependent and independent variables used in cover preference regression model.**

| Dependent Variables | Variable Descriptions |
|---|---|
| Woody cover preference | Categorical variable response to question, "Which proportion of trees/shrubs to grassland would you prefer on your property?": approximately 5%, approximately 25%, approximately 50%, approximately 75% |
| **Independent Variables** | |
| PBA membership | Binary single item variable (0 = No, 1 = Yes) |
| State of property location | Binary single item variable (0 = Oklahoma, 1 = Texas) |
| Gender | Binary single item variable (0 = female, 1 = male) |
| Age (years) in 2015 | Continuous single item variable |
| Education level | Ordinal variable: High school (reference category), some post-secondary/ bachelor degree, graduate/professional degree. |
| Property size | Ordinal variable: 40.5–202 ha (100–500 acres = small, reference category), 203–404 ha (501–1000 acres = medium), > 404 ha (>1000 acres = large) |
| Period of residence on property | Ordinal variable: absentee resident (reference category), occasional resident, and full time resident |
| Income from property | Ordinal response for, "In 2014, approximately what percent of your total annual income was generated from activities on your rural property?" 0% (reference category), 1–25%, 26–50%, 51–75%, 76–100%. |
| Lifestyle/recreation as ownership motivator | PCA derived latent variable (answer scale ranging in value from 1 = not at all important to 7 = very important) |
| Ranching as ownership motivator | PCA derived latent variable (answer scale ranging in value from 1 = not at all important to 7 = very important) |
| Heritage as ownership motivator | PCA derived latent variable (answer scale ranging in value from 1 = not at all important to 7 = very important) |
| Hunting as ownership motivator | PCA derived latent variable (answer scale ranging in value from 1 = not at all important to 7 = very important) |
| Farming as ownership motivator | PCA derived latent variable (answer scale ranging in value from 1 = not at all important to 7 = very important) |

overall 36.7% useable response rate. However, response rates differed by state and by category of survey participants–general landowners versus PBA member landowners (Table 3). Overall, the response rate was higher in Texas than in Oklahoma and PBA members were much more likely than general landowners to participate in this study.

The Mann-Whitney rank tests conducted to determine if there was potential response bias among PBA members compared to general respondents found no statistically significant differences between these two groups of respondents with respect to the proportion of annual income derived from the property or about existing and preferred woody plant density (Table 4). However, PBA members were more likely to respond that woody plants are a problem on their land (non -members also agreed with that statement but not as strongly), and they were much more likely to favor the application of prescribed fire to their own land (both were p < 0.0001).

Additionally, Mann-Whiney tests used to compare early and later respondent data identified no statistically significant differences between the two groups with respect to age of respondent, reliance on income from their property, perception of and preference for woody plant cover on their land, and whether or not they thought woody plants were a problem on their land (p = 0.0707–0.7721).

## Landowner characteristics

Respondents were predominantly older (average age of 66 years), male (81%) and well educated (85% reported at least some college) (Table 5). Over half of the respondents (57%) lived

**Table 2. Orthogonal varimax rotated factor loading results of PCA analysis of landownership motivations with Cronbach's α measuring internal scale reliability.**

| Landownership motivations | Lifestyle/ recreation α = 0.8574 | Ranching α = 0.8526 | Heritage α = 0.9206 | Hunting α = 0.6289 | Farming α = 0.6693 | Mean response score[a] |
|---|---|---|---|---|---|---|
| Place to relax | **0.8883** | 0.0195 | 0.0572 | 0.0747 | 0.0117 | 6.18 |
| Enjoy the outdoors | **0.8132** | 0.1247 | 0.1615 | 0.0798 | -0.0706 | 6.51 |
| Hunting/fishing (recreational) | **0.8020** | -0.0949 | 0.0590 | 0.2146 | 0.0697 | 5.89 |
| Non-hunting/fishing recreation | **0.8386** | 0.0450 | -0.0060 | 0.2309 | 0.0412 | 5.89 |
| Operate a farm/ranch | 0.0702 | **0.8709** | 0.1704 | 0.0710 | 0.0980 | 6.15 |
| Maintain family farming/ranching tradition | 0.0004 | **0.6943** | 0.5299 | 0.0843 | 0.1167 | 5.92 |
| Produce grazing livestock | 0.0185 | **0.8561** | 0.1962 | 0.0040 | 0.0942 | 6.06 |
| Earn a profit | -0.0270 | **0.6702** | 0.1280 | 0.0774 | 0.3765 | 5.68 |
| Keep land in family | 0.0669 | 0.2411 | **0.917** | 0.0169 | 0.0608 | 6.43 |
| Leave land for family | 0.1050 | 0.1664 | **0.918** | 0.0655 | 0.0738 | 6.42 |
| Operate a hunting enterprise | 0.0177 | 0.2631 | 0.0877 | **0.8071** | -0.0769 | 3.90 |
| Manage large wildlife (deer) | 0.2923 | -0.0128 | -0.0010 | **0.8657** | -0.0793 | 4.82 |
| Manage game birds | 0.2770 | -0.0773 | 0.0799 | **0.7387** | 0.1939 | 4.62 |
| Produce hay/forage | 0.0599 | 0.3569 | 0.1370 | -0.2315 | **0.6425** | 4.63 |
| Cultivate crops | -0.0216 | 0.1995 | 0.1282 | -0.0159 | **0.7628** | 3.61 |
| Obtain income from minerals | -0.0988 | 0.0775 | 0.1841 | 0.1023 | **0.6747** | 4.02 |
| Have a financial investment | 0.2230 | 0.1096 | -0.1402 | 0.0359 | **0.6175** | 5.49 |
| Eigenvalue | 4.6 | 3.4 | 1.7 | 1.5 | 1.1 | - |
| % explained variance | 17.9 | 16.5 | 12.8 | 12.6 | 12.2 | - |

Shaded results indicate variables that load on a particular factor.

[a] Mean response scores based on Likert scale 1 = not at all important to 7 = very important.

in Texas and over half (54%) lived on their rural land full time. Approximately equal proportions of respondents owned properties that were 45.5–404 ha (100–1,000 ha) and > 405 ha (1,000 acres) in size. The large majority (84%) of the respondents reported earning some of their annual income from their rural land, and over a quarter (28%) reported they earned more than half of their income from activities on their rural land. About a third of respondents (32%) reported being a member of a PBA.

## Woody plant cover perceptions and preferences

When we asked participants if cedar or juniper had increased on their property in the last ten years, 68% responded that it had (36% strongly agreed) and 52% of respondents agreed that mesquite had increased on their land in the last ten years (24% strongly agreed). However, we also asked participants to estimate their current and preferred amount of woody plant cover based on a visual schematic (Fig 2). Fig 3 presents our findings about how the survey

**Table 3. Survey questionnaire response rates.**

| Response Rates by group | Responses received | % of total respondents | Intra-group response rate |
|---|---|---|---|
| Texas general landowners | 272 | 40.0 | (272/800) = 34% |
| Texas PBA members | 112 | 16.5 | (112/126) = 89% |
| Oklahoma general landowners | 192 | 28.2 | (192/800) = 24% |
| Oklahoma PBA members | 104 | 15.3 | (104/192) = 54% |
| Total | 680 | 100% | |

**Table 4. Mean response scores from PBA members and non-member survey questionnaire responses testing non-random participation bias.**

| Survey question | PBA mean (n = 216) | Non-PBA mean (n = 464) | % diff. in means | Mann-Whitney sig. |
|---|---|---|---|---|
| % annual income earned from property [a] | 2.9 | 2.6 | 11.5 | 0.0177 |
| Current proportion of woody plant to grassland on property [b] | 2.7 | 2.6 | 3.8 | 0.3289 |
| Preferred proportion of woody plants to grassland on property [b] | 1.7 | 1.9 | 11.7 | 0.0771 |
| Woody plants are a problem on my land [a] | 6.3 | 5.7 | 9.5 | **<0.0001** |
| In favor of applying prescribed fire on my land [a] | 6.5 | 4.9 | 32.7 | **<0.0001** |

[c] Mean response score based on "income from property" variable categories listed in Table 1.

[b] Mean response score based on "woody cover preference" variable categories listed in Table 1.

[a] Mean response scores based on Likert scale 1 = strongly disagree to 7 = strongly agree.

respondents' perceptions about woody cover on their land compared with their preferred amount of woody cover. In Texas, 80% of respondents estimated they had 50% or more woody plant cover on their land, whereas 67% of the Oklahoma respondents reported they had 25% or less woody plant cover. In both states, the large majority of respondents preferred less woody cover than they estimated they had on their land. Overall, 78% of Texas respondents

**Table 5. Survey respondents' demographic and property characteristics.**

| *Demographic variable* | *Survey respondents* |
|---|---|
| **Age** | Range = 30–93, mean = 66 years, std. dev. = 11 |
| **Gender** | |
| Male | 81% |
| Female | 19% |
| **Education** | |
| Some high school/high school graduate | 15% |
| Some college/college graduate | 50% |
| Some graduate school/graduate degree | 35% |
| **State of residence** | |
| Texas | 57% |
| Oklahoma | 43% |
| **Residency on rural property** | |
| Full time resident | 54% |
| Part-time/weekend resident | 19% |
| Non-resident | 27% |
| **Property size** | |
| 40.5–202 ha (100–500 acres–small) | 29% |
| 203–404 ha (501–1000 acres–medium) | 20% |
| > 406 ha (>1000 acres–large) | 51% |
| **Annual income from property** | |
| 0% | 16% |
| 1–25% | 40% |
| 26–50% | 16% |
| 51–75% | 14% |
| 76–100% | 14% |
| **PBA member (self-identified)** | |
| Yes | 32% |
| No | 68% |

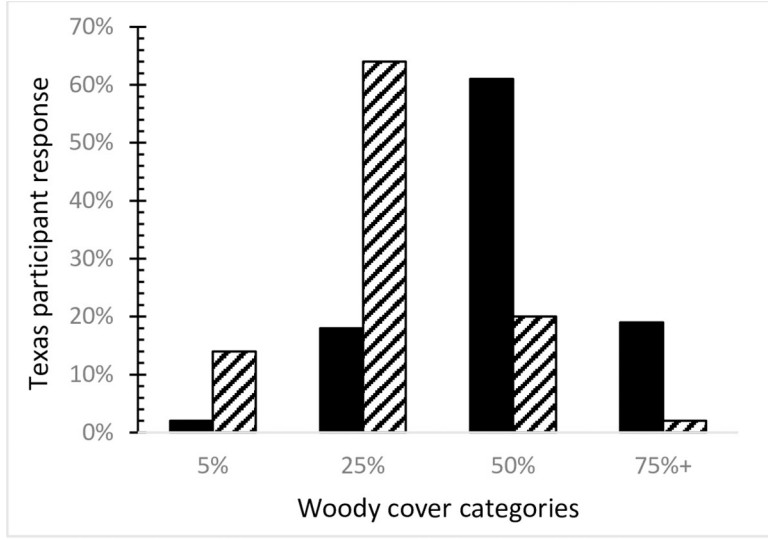

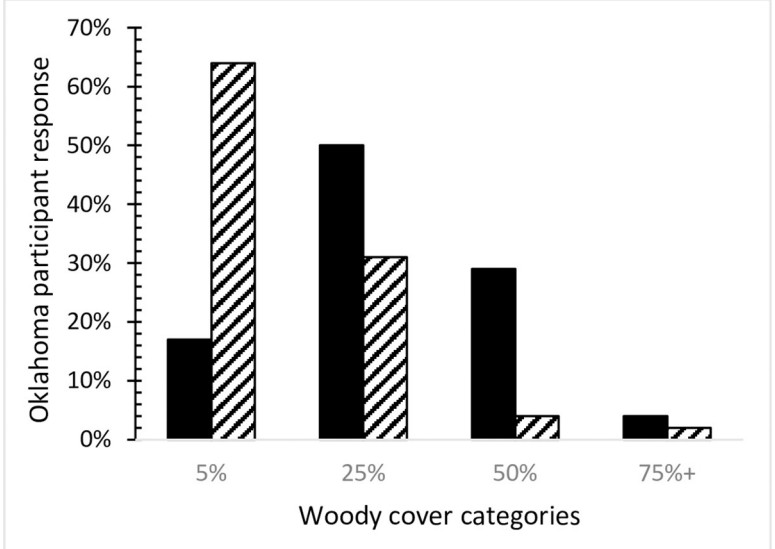

**Fig 3. Comparison of Texas and Oklahoma respondents' perceptions of current woody plant cover on their land (black bars) with their preferred amount of woody plant cover (striped bars).**

and 94% of Oklahoma wanted woody cover on their land to be 25% or less, but the predominantly preferred woody cover was 25% in Texas (64% of respondents) and 5% in Oklahoma (64% of the respondents). Therefore, in both states, respondents preferred less woody plant cover but the Oklahoma respondents expressed a stronger preference for more open grassland.

Fig 4 presents the degree of separation between each respondent's preferred and estimated woody plant coverage. A zero value indicates those respondents whose preferred and estimated coverage were the same, i.e., they are satisfied with their current amount of woody plants, whereas negative values increasingly indicate respondents' desire for less cover then they currently have. Approximately 69% of respondents indicated that they preferred less woody cover with over half desiring a 25% reduction in woody cover.

We were also interested in how respondents' perceptions of woody plant cover change matched with satellite data measuring cover in the sampled counties. We were able to obtain

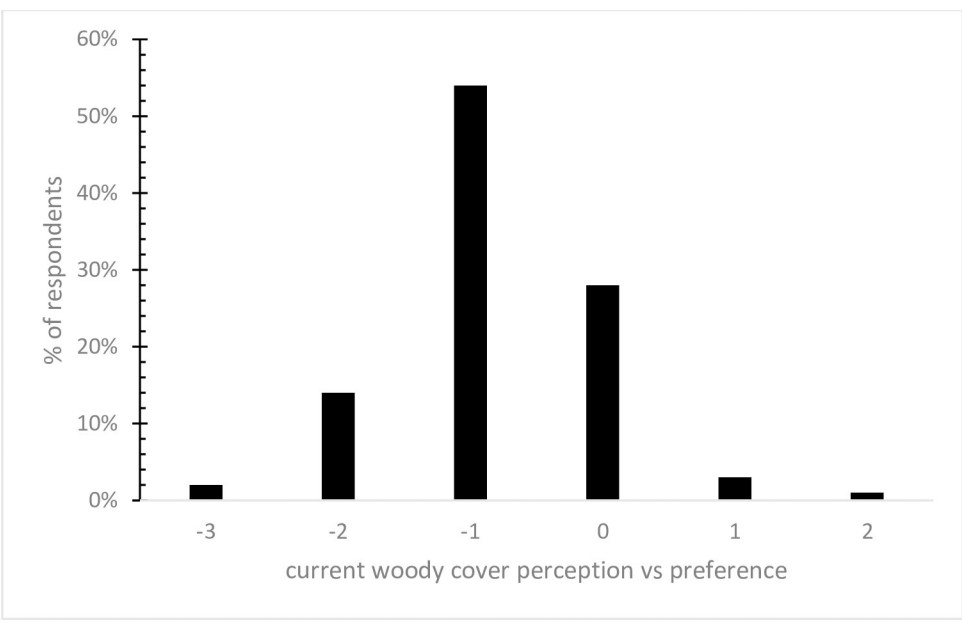

**Fig 4. Difference between respondent's preference for woody plant cover and the current amount of cover on their land.** Values < 0 indicate preference for less woody plant cover.

Landsat data measuring woody plant cover increases or decreases between 2004–2014 in our sampled counties, some of which are available in the published literature [14]. While the Landsat data, presented in Table 6, reflects entire counties and not the respondents' specific

**Table 6. Landowner estimates of woody cover increases and documented woody cover increases by county between 2004–2014.**

| State/County | Approximate woody plant cover % 2004 2014 | % change in woody cover within county 2004–2014[a] | % of respondents who perceived of woody plant increases in their county [b] |
|---|---|---|---|
| **Texas** | | | |
| Schleicher | 17.3% 20.0% | **2.8%** | 78% |
| San Saba | 25.1% 25.2% | **0.1%** | 79% |
| Menard | 24.3% 23.5% | -0.8% | 69% |
| Sutton | 26.0% 23.0% | -2.9% | 88% |
| Gillespie | 35.9% 32.6% | -3.4% | 76% |
| Llano | 27.9% 23.3% | -4.6% | 71% |
| Mason | 28.1% 22.6% | -5.5% | 54% |
| Kimble | 34.8% 24.4% | -10.4% | 87% |
| **Oklahoma** | | | |
| Pawnee | 24.7% 36.3% | **11.6%** | 80% |
| Payne | 25.3% 36.5% | **11.3%** | 78% |
| Beckham | 5.4% 9.0% | **3.6%** | 69% |
| Dewey | 11.0% 14.6% | **3.6%** | 87% |
| Roger Mills | 5.4% 8.8% | **3.4%** | 77% |
| Comanche | 10.7% 13.7% | **3.0%** | 73% |
| Tillman | 3.7% 4.9% | **1.2%** | 35% |
| Ellis | 5.1% 4.4% | -0.7% | 70% |

[a] Bolded values indicate increasing woody cover in that county between 2004–2014 [15]
[b] % of respondents within county indicating increased woody cover in last 10 years

properties, it does serve to highlight some interesting dichotomies. The majority (>50%) of respondents in all but Tillman county in Oklahoma perceived that woody plant cover increased on their land during the preceding 10 years, and in nine of the 16 counties in the study more than 75% of the respondents believed this to be the case (Table 6). However, in 2018 Hartfield and van Leeuwen reported that, based on Landsat data, only nine of these counties including seven in Oklahoma reflected woody plant increases while six of the counties in Texas experienced a 1–10% decrease during that time period [15].

## Regression analyses

The ordinal logistic regression model for factors influencing respondents' preference for woody plant cover is presented in Table 7. The most common drivers of cover preference pertain to landownership motivations. Respondents who owned their land for lifestyle or recreational purposes and those who owned their land for hunting purposes were significantly more likely to prefer more woody plant cover, while those who owned their land for ranching purposes were more likely to prefer less woody plant cover and more grassland for their livestock. In addition, the regression model emphasized that Texas respondents preferred a greater amount of woody cover then Oklahoma respondents. Older respondents were also slightly more likely to prefer less woody cover on their land.

Finally, we wanted to determine if woody cover preference was correlated with willingness to use prescribed fire as a relatively low cost, effective tool for reducing woody plants (Table 8) [9]. The regression analysis results found no link between cover preference and use of prescribed fire. Respondents from Oklahoma were about 61% more likely than Texas respondents to have used prescribed fire on their land indicating a stronger pro-fire culture in Oklahoma,

**Table 7. Ordinal logistic regression model for factors influencing woody plant cover preferences of respondents in Texas and Oklahoma.** (Bold items indicate P <0.05).

| Independent Variables | Woody cover preference model: Pseudo $R^2$ = 0.206; P<0.001 | | | |
|---|---|---|---|---|
| | β coeff. | Std. err. β coeff. | p-value | % Δ odds |
| **Lifestyle/recreation motivation** | **0.179** | 0.095 | **0.050** | **19.7** |
| **Rancher motivation** | **-0.442** | 0.111 | **<0.001** | **-34.5** |
| Heritage motivation | -0.151 | 0.089 | 0.091 | -14.1 |
| **Hunter motivation** | **0.492** | 0.112 | **<0.001** | **63.7** |
| Farmer motivation | -0.178 | 0.108 | 0.101 | -16.3 |
| Occasional resident [a] | 0.306 | 0.274 | 0.265 | 35.8 |
| Full time resident [a] | -0.024 | 0.231 | 0.919 | -2.3 |
| **State** | **1.761** | 0.245 | **<0.001** | **482.0** |
| PBA member | -0.186 | 0.204 | 0.360 | -17.0 |
| Income = 1–25% [b] | 0.205 | 0.299 | 0.493 | 22.8 |
| Income = 26–50% [b] | -0.387 | 0.390 | 0.322 | -32.1 |
| Income = 51–75% [b] | -0.114 | 0.413 | 0.783 | -10.8 |
| Income = 76–100% [b] | -0.572 | 0.433 | 0.186 | -43.6 |
| Large sized properties [c] | -0.196 | 0.252 | 0.439 | -17.8 |
| Medium sized properties [c] | 0.292 | 0.271 | 0.283 | 33.9 |
| Gender | -0.404 | 0.243 | 0.096 | -33.2 |
| **Age in 2015** | **-0.018** | 0.008 | **0.029** | **-1.9** |

[a] Absentee resident is reference category

[b] 0% of annual income derived from rural property is reference category

[c] small property (40.5–202 ha, 100–500 acres) is reference category

**Table 8. Logistic regression results on factors influencing use of prescribed fire on private lands.** (Bolded results indicate P <0.05).

| Independent Variables | Use of prescribed fire model: Pseudo R$^2$ = 0.209; P<0.001 | | | |
|---|---|---|---|---|
| | β coeff. | Std. err. β coeff. | p-value | % Δ odds |
| Woody cover preference | 0.167 | 0.165 | 0.313 | 18.2 |
| Lifestyle/recreation motivation | 0.133 | 0.106 | 0.212 | 14.3 |
| Rancher motivation | 0.005 | 0.134 | 0.973 | 0.5 |
| Heritage motivation | -0.035 | 0.109 | 0.752 | -3.4 |
| Hunter motivation | 0.004 | 0.121 | 0.975 | 0.4 |
| Farmer motivation | -0.032 | 0.123 | 0.796 | -3.1 |
| **State** | **-0.963** | 0.278 | **0.001** | **-61.8** |
| **PBA member** | **2.169** | 0.238 | **<0.001** | **774.6** |
| Income = 1–25% [a] | 0.314 | 0.360 | 0.383 | 36.9 |
| Income = 26–50% [a] | 0.558 | 0.442 | 0.207 | 74.8 |
| Income = 51–75% [a] | 0.422 | 0.468 | 0.368 | 52.6 |
| Income = 76–100% [a] | 0.757 | 0.471 | 0.109 | 113.1 |
| **Large sized properties [b]** | **0.657** | 0.284 | **0.021** | **92.8** |
| Medium sized properties [b] | 0.195 | 0.315 | 0.536 | 21.6 |
| Gender | 0.153 | 0.270 | 0.569 | 16.6 |
| **Age in 2015** | **-0.025** | 0.009 | **0.008** | **-2.5** |

[a] 0% of annual income derived from rural property is reference category

[b] small property (40.5–202 ha, 100–500 acres) is reference category

and respondents in both states who owned large sized properties (>406 ha, 1000 acres), were also significantly more likely to use fire than landowners with smaller properties. Unsurprisingly, PBA members were significantly more likely (774%) to have used prescribed fire on their property than non-members. Finally, older landowners were significantly less likely to have used prescribed fire.

## Discussion

The survey participants in the Edwards Plateau and the Rolling Plains study areas in Texas and Oklahoma overwhelmingly reported increases in woody plants on their property. However, based on the additional remote sensing data from the sampled counties, only 9 of the 16 counties actually had overall woody plant increases, while six counties experienced decreases in overall woody plant cover during the 10 years preceding this survey.

The disparity between respondents' perspectives about woody plant cover and the results of the land cover change analysis is striking. It is possible that individual landowners are experiencing woody plant increases but it is unlikely that significant woody plant expansion has occurred on most properties during that 10-year time period. There are several potential reasons behind the seeming disconnect between perception and documented land cover changes. The land cover change estimates may have been influenced by the effects of a severe drought between 2000 and 2011 in that dead standing trees, which were more abundant in the Edwards Plateau, may have been undetected from space whereas landowners could easily see them from ground level [19–21]. Landowners are also able to see small saplings that indicate woody plant increase but these may also not be detectable from space. Additionally, Roberts et al. (2018) discovered that nearly all policies aimed at containing juniper tend to promote rather than contain the spread of the species due to mismatches between policy and ecology and contradictory programs within natural resources agencies that simultaneously promote

and control invasive species. [22] Therefore, reasons why ground level perceptions of woody plant expansion may be inconsistent with estimates derived from remotely-sensed data need to be explored in order to formulate meaningful and scientifically accurate policy guidelines for managing woody plant expansion effectively [4]. Developing simple tools that landowners can use to quantify woody plant cover on their property may also be useful for accurately monitoring changes over time. If landowners do not have an accurate representation of the extent of woody plant expansion on their land, they may not perceive woody plants as a problem.

Our study also found significant inter-state differences in woody cover preferences, with Texas respondents generally preferring higher cover (25%) compared to their Oklahoma counterparts (5%). In part, the preference for more woody cover among Texas respondents may be due to the marked increase in Central Texas of landowners who are moving away from traditional farming and ranching into recreational or wildlife-centric land uses [23,24]. By contrast, the lower woody cover preference in Oklahoma may be attributable to the continued importance of the ranching industry, particularly cattle ranching [25]. One question our research did not address is what type and patterns of woody plant cover do landowners prefer. Previous research found that many landowners within the Southern Plains demonstrate a preference for a moderately heterogeneous grassland landscape whereas those who were more production focused tended not to prefer highly heterogeneous landscapes [26]. This complexity of preferences among landowners may also extend to areas of woody plant encroachment or mixed grassland/woodland landscapes. For example, some landowners may want to maintain preferred tree species, such as mature oaks (*Quercus sp.*), while other may prefer trees that provide shade, cover, supplemental food or windbreaks for livestock or wildlife species [27,28].

Our results indicated that landownership motivations are significant drivers of woody cover preference, a finding corroborated by others [23,24]. Those who owned their property for recreation or hunting preferred more woody plant cover while more traditional livestock-oriented ranchers preferred less. This finding highlights the need for continuing research into changing land use trends. If regions experience a shift from larger livestock-oriented ranches to smaller properties that are owned primarily for lifestyle and recreational purposes, then woody cover expansion is likely to continue, in part, because the latter landowners tend to prefer more woody plants on their land [1,24]. This has serious ramifications for the maintenance or restoration of open grasslands and their associated biodiversity. It may also escalate wildfire risk in rural areas with increasing woody cover and development, which is typical of ever larger portions of the Edwards Plateau in Texas. Increasing risk of wildfires is of particular concern in areas where cedar and juniper species are dominant, as they are considered volatile fuel sources [4,29].

Given our finding that most of our respondents preferred lower woody plant density on their land than they currently have and the fact that prescribed fire has been shown to be an ecologically effective and economical efficient management tool to contain woody plant expansion [4,9], we anticipated there would be a negative correlation between preferred woody plant cover and use of prescribed fire. However, while our study did find that Oklahoma respondents who preferred lower woody plant cover appeared to be more accepting of prescribed fire than the Texas respondents, our study did not find any statistically significant correlation between woody cover preference and the use of prescribed fire on private lands. One reason for this is likely that many landowners still do not consider fire a safe and effective tool for managing rangelands [11]. Furthermore, encouraging the use of fire on private lands has proven to be complicated for a variety of reasons including landowners perceived risk from fire both as a physical hazard and from a legal liability standpoint [11,12,30,31] and a lack of knowledge or confidence in application of fire [6]. This suggests the urgent need for outreach programs aimed at educating landowners about the ecological and economic efficacy

of periodic prescribed fire for containing and reducing woody plant cover and that also provide training and support in the use of prescribed fire to reduce concerns about the legal liability of using this management tool.

However, previous research has also shown that while educational programs can increase landowner's knowledge and tolerance of prescribed fire programs, that may not translate into adoption of the practice [32]. By contrast, it has been found that PBA's are very effective in engaging landowners in the use of prescribed fire, because they provide their members with fire safety training and equipment and they promote safe fire application experience among members who assist each other on burn days [4,33–35]. Other research has also supported the use of social networks in order to achieve land management objectives, especially in increasing landowners access to labor resources [36]. Given that our research also found that one of the differences between PBA members and non-PBA members was that PBA members were more likely to view woody plants on their land as a problem, PBA membership may be the catalyst for the greater use of prescribed fire to combat woody plant expansion. Therefore, in order to use prescribed fire to reduce woody fuel accumulations and thereby mitigate the increasing risk of wildlife, one important avenue for future research is how to best support and encourage the growth of PBA's.

We also found age to be a significant factor both in woody cover preference and use of prescribed fire. Our average respondent age was 66 years old, highlighting the fact that many rural landowners tend to be older, which is a common finding in rural landowner surveys in our study area and throughout the US. Our results showed that older landowners may be slightly less likely to use prescribed fire but other research has indicated that the ages of the people who actually work on a farm or ranch is a better predictor of applied land management than the age of the primary decision maker [6,37–39].

While we feel that this study provides important preliminary evidence relating to landowners' views about woody plant expansion and prescribed fire in the Southern Plains region of the United States, there are some limitations that should be addressed in future research. First, the purposive inclusion of all PBA members in our study area versus use of a random sample of non-member landowners may have produced biased results, a concern we attempted to remedy by comparing PBA member and non-member data relating to key study questions. While those tests found no difference in terms of perception or preference of woody plants on their land, PBA members were more likely to view woody plants as a problem and, not surprising, have a more positive view of prescribed fire. However, the directionality of this difference is unclear; i.e., whether PBA membership elevated the perception of woody plants as a problem or if landowners who viewed woody plants as a problem were more likely to become a PBA member in order to apply fire using PBA membership benefits including access to labor, fire management equipment and fire training. Future research should endeavor to identify changes in landowner perspectives about woody plant density and prescribed fire before and after joining a PBA. Second, we could not follow up with non-respondents and, therefore, were unable to rigorously ascertain if there was any non-response bias in our data. Others have recommended comparing data from early versus late respondents for testing non-response bias [14]. Our own comparative tests did not find any statistically significant differences between the two groups. Nevertheless, given that we found no evidence that this approach can accurately identify non-response bias, we opted to limit the discussion of our results to the respondents rather than extrapolate out findings to the general landowner population from which the sub-sample was drawn. Future research of landowner perspectives of woody plant densities and prescribed fire should endeavor to obtain a substantially higher survey response rates and some comparable data from non-respondents.

## Conclusion

This study sought to address three research questions. 1) Do landowner perceptions about woody plant expansion correspond with estimated changes in their area? 2) Do landowners within the Southern Plains desire less woody cover on their land? 3) Do expressed woody cover preferences influence the adoption of prescribed fire on private land? Our results indicated that there was a disconnect between the perception of woody plant expansion on private lands and actual, local measurements of woody plant cover. Specifically, many landowners indicated that woody plants had increased on their own properties during the preceding 10 years whereas county level measurements showed little change in overall woody cover. Within the Texas and Oklahoma study areas, the survey respondents generally preferred about 25% less woody plant cover on their land than they estimated they have. Given the efficacy of prescribed fire as a woody plant management tool, we expected a positive correlation between degree of dissatisfaction with woody plant cover and willingness to use this management tool to combat woody plant expansion but this expectation was not confirmed in our study. These findings highlight the complexity of determining the relationship between landowner management goals, perceptions of woody plant cover and prescribed fire. This suggests the need for effective outreach programs to make the connection between woody plant reduction and prescribed fire for diverse and shifting types of landowners in the Southern Plains and to increase the number of prescribed burn associations across the ecoregion to encourage more landowners to use this management tool to restore grasslands and reduce wildfire risks on their land.

## Supporting information

**S1 File. Mail survey questionnaire.**
(PDF)

**S2 File. Anonymized data file.**
(XLS)

## Acknowledgments

The authors want to thank all 1853 landowners who participated in this study.

## Author Contributions

**Conceptualization:** Dianne A. Stroman, Urs P. Kreuter.

**Data curation:** Dianne A. Stroman.

**Formal analysis:** Dianne A. Stroman, Urs P. Kreuter, Carissa L. Wonkka.

**Funding acquisition:** Urs P. Kreuter.

**Investigation:** Dianne A. Stroman, Urs P. Kreuter.

**Methodology:** Dianne A. Stroman, Urs P. Kreuter.

**Project administration:** Dianne A. Stroman, Urs P. Kreuter.

**Resources:** Urs P. Kreuter.

**Supervision:** Urs P. Kreuter.

**Writing – original draft:** Dianne A. Stroman, Urs P. Kreuter.

**Writing – review & editing:** Dianne A. Stroman, Urs P. Kreuter, Carissa L. Wonkka.

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
