## [Decision Letter · Decision Letter 0]

24 Oct 2019

PONE-D-19-20616

Landowner perceptions of woody plants and prescribed fire in the Southern Plains, USA

PLOS ONE

Dear Dr. Stroman,

Thank you for submitting your manuscript to PLOS ONE. After careful consideration, we feel that it has merit but does not fully meet PLOS ONE’s publication criteria as it currently stands. Therefore, we invite you to submit a revised version of the manuscript that addresses the points raised during the review process.

ACADEMIC EDITOR: The authors address a topic of potential interest not just in the US but in other regions around the world facing similar challenges and opportunities. Reviewers have raised various issues that shall be properly addressed to merit the publication of this manuscript.

We would appreciate receiving your revised manuscript by Dec 08 2019 11:59PM. To enhance the reproducibility of your results, we recommend that if applicable you deposit your laboratory protocols in protocols.io, where a protocol can be assigned its own identifier (DOI) such that it can be cited independently in the future. For instructions see: http://journals.plos.org/plosone/s/submission-guidelines#loc-laboratory-protocols

We look forward to receiving your revised manuscript.

Kind regards,

Francisco X Aguilar

Academic Editor

PLOS ONE

Journal Requirements:

2.  Please include additional information regarding the survey or questionnaire used in the study and ensure that you have provided sufficient details that others could replicate the analyses. For instance, if you developed a questionnaire as part of this study and it is not under a copyright more restrictive than CC-BY, please include a copy as Supporting Information.

3. 

We note that you have indicated that data from this study are available upon request. PLOS only allows data to be available upon request if there are legal or ethical restrictions on sharing data publicly. For more information on unacceptable data access restrictions, please see http://journals.plos.org/plosone/s/data-availability#loc-unacceptable-data-access-restrictions.

4. 

We note that Figures 1 in your submission contain [map/satellite] images which may be copyrighted. All PLOS content is published under the Creative Commons Attribution License (CC BY 4.0), which means that the manuscript, images, and Supporting Information files will be freely available online, and any third party is permitted to access, download, copy, distribute, and use these materials in any way, even commercially, with proper attribution. For these reasons, we cannot publish previously copyrighted maps or satellite images created using proprietary data, such as Google software (Google Maps, Street View, and Earth). For more information, see our copyright guidelines: http://journals.plos.org/plosone/s/licenses-and-copyright.

a)   You may seek permission from the original copyright holder of Figure(s) [#] to publish the content specifically under the CC BY 4.0 license. 

Reviewers' comments:

Reviewer's Responses to Questions

**Comments to the Author**

1. Is the manuscript technically sound, and do the data support the conclusions?

Reviewer #1: Partly

Reviewer #2: Yes

Reviewer #3: Partly

2. Has the statistical analysis been performed appropriately and rigorously? 

Reviewer #1: Yes

Reviewer #2: Yes

Reviewer #3: Yes

3. Have the authors made all data underlying the findings in their manuscript fully available?

Reviewer #1: No

Reviewer #2: No

Reviewer #3: No

4. Is the manuscript presented in an intelligible fashion and written in standard English?

Reviewer #1: No

Reviewer #2: Yes

Reviewer #3: Yes

5. Review Comments to the Author

Reviewer #1: Review of manuscript PONE-D-19-20616 – Landowner perceptions of woody plants and prescribed fire in the Southern Plains, USA

Summary

This study reviewed attitudes of landowners in two geographic regions of western Oklahoma and south-central Texas for comparing perceived and preferred woody plant cover and how these preferences relate to use of prescribed fire. Respondents indicated that they had more woody plant cover on their property than desired. Recreational and hunter landowner categories were more likely to prefer woody cover, while the rancher category was less likely, indicating that primary ownership motivations can influence woody cover preference.

Attitudinal research such as this is a beneficial tool for revealing regional motivations of private landowners and can help guide future management/conservation approaches while furthering the institutional knowledge of the subject. The authors used good survey methodology, including an easy to understand visual graphic for estimation of woody cover. The use of various forms of data analyses helped to illustrate and support findings, and there was good categorical class delineation for the latent variables of primary ownership motivators, including testing for internal reliability. The authors also did a good job of providing relevance for the study by discussing relationship to a larger perspective.

However, there were noticeable areas for improvement before the acceptance of the manuscript. This includes need for discussion about how potential sources of bias were handled, manuscript organization, and a focus on incorporating research data into the Discussion and Conclusion section. Several minor issues will also need to be addressed.

Examples/Evidence

Major Issues

1) Discuss how potential bias was handled

a. Survey recipients included 100 random landowners from each county in the study area, plus PBA members. It seems that including a specific subset of the sample population would potentially influence findings and make it difficult to draw conclusions, especially since PBA members would already understand the value of prescribed fire and therefore are potentially more aware of the presence of woody plants on their property. How was this potential source of bias accounted for during the analyses?

b. Were there any other sources of bias tested for? (e.g. violations of assumptions or outliers)

c. How were unanswered questions handled? (i.e. was there any influence created by non-responses to specific items within questionnaire areas of inquiry)

2) Manuscript organization

a. Information in the Results section should only report results; discussion should be located in the Discussion section (line 240-243)

b. Data should be reported in the Results section (Table 7)

3) Results relationship to Discussion and Conclusions

a. It is important to use the Discussion and Conclusion section to show how the study complements the existing literature and answers the research questions. The authors do a good job of touching on highlights, but more of this study’s research findings should be listed, as well as referencing tables and figures from the Results sections.

Minor Issues

4) Data/analyses transparency

a. What software package was used for data analyses?

b. Insert table to show questionnaire items that composed attitudes concerning woody plants and fire. This would provide context on the number and type of questions defining each construct and would be an opportunity to indicate response %. Also indicate which items were positively and negatively worded for purpose of reverse-scoring in the data analyses.

c. Table 2

i. What type of rotation was used?

ii. List rotated factor loading results for all items

iii. Include % of Variance as a separate row

d. Table 5 and 6

i. Include standard errors for β coeff.

ii. Consider reporting odds ratio and confidence interval instead of % Δ odds, as this will help with interpretation of the results

5) Include supporting references for statements

a. line 284

b. line 296-7 – in that area? statewide?

c. line 315

d. line 241-3 – include reference; how does this relate to Table 4?

e. line 315-6

6) Clarity of statements

a. line 250, 254 – be careful of interpreting % Δ odds as probability; reporting the odds ratio will allow for more accurate interpretation of data

b. line 282-3 – what about the possible influence of PBA member bias?

c. line 286 – expand on this thought

d. line 303-5 – how does this species relate to the study, or include reference for why it should be specifically mentioned

e. line 314 – discuss how/why would it escalate wildfire risk

f. line 344 – what preceding questions? If they are the research questions then they should be restated

7) Completeness of reporting information

a. Be sure to discuss all statistically significant findings – ex. age relationship and woody cover preference (Table 5) – all other significant findings were mentioned, including age for Table 6

8) Proofreading

a. Several typos and grammatical errors exist and need corrected

Reviewer #2: Well conceived and well written paper addressing a topical public policy issue focusing on social drivers of woody cover change in the US Great Plains. The strength of the analysis rests on the stratified random selection of respondents. However, in lines143-148 you note that you do not conduct an assessment of cover change on respondent's properties--a weakness--but you adequately attempt to qualify your results / observations. L120 has an extra 'to', and presumably L129 is missing a 'not'

Reviewer #3: The authors conducted a mail survey in sixteen counties in two states to determine (1) how landowners perceived the current woody cover of their land and whether it had changed over the last 10 years, (2) What their desired woody cover was and how this compared to their perception of actual woody cover, and (3) whether an expressed preference for lower woody cover was linked to a land owner being more likely to have used fire in the past. The responses were analysed to determine whether factors such as landowner demographic characteristics, type of land use, property size and membership of a prescribed burning association (PBA) influenced woody cover preference and use of prescribed burning. The authors make the case that woody cover increase is a significant ecological and economic problem in the study area, and that knowing about different woody cover preferences and factors that influence adoption of prescribed burning will help target appropriate interventions to improve the use of fire to control woody encroachment.

It seems odd to conduct this research on issues surrounding woody cover increase in counties that actually have had decreasing woody cover (Table 7). Was there any baseline data on woody cover and woody cover change in the study area that informed the study and choice of study sites?

The study appears to be largely methodologically sound, but I have three main concerns that need to be addressed. First, the description of the survey and the questionnaire is insufficient to judge the study design and to allow for repeatability. Second, the possible biases introduced the survey methods are incompletely acknowledged and should be dealt with more explicitly. Third, the inclusion of data from a remote sensing study to contextualize the perceptions expressed by landowners in the survey is a good idea, but it was poorly done.

1. Incomplete information on the questionnaire

It would have been useful to have access to the whole (blank) questionnaire. This would allow readers to assess how the questions were phrased, and whether some information that would have been useful to have (e.g. reasons given for using/not using prescribed burning) was collected but not reported, or whether it was not collected. It would also allow for repeatability of the work.

The methods sections states that “we asked the survey participants a series of questions as part of a broader mail survey about woody plants and prescribed fire” (lines 107-108) and the questionnaire was apparently 12 pp long (line 116). This suggests that there was material covered in the questionnaire that is not reported in this manuscript, but which may be useful to include to aid interpretation. Are there substantial aspects of the questionnaire that are not reported in this manuscript? If so, what are the reasons for this? I feel that I am getting an incomplete picture of the questionnaire and its overall aims. Are they the same as reported in this paper or is the work presented here just a subset?

The questionnaire asked respondents to estimate the current woody cover on their land and to indicate their preferred amount of woody cover. Farmers were also asked whether they had perceived an increase in woody cover. It would have been useful to have asked formers to estimate the woody cover on their farms at some earlier data using the sample schematics used to elicit perceptions of present and desirable woody cover.

The authors state that the survey data cannot be freely shared because of ethics/confidentiality concerns, but this restriction should not apply to the actual questionnaire that was used.

2. Potential sampling bias resulting from the study design

The authors simply state that “A non-response bias analysis was not conducted because alternative contact information (e.g., telephone number) required for such an analysis was not obtainable. Due to this limitation, we do extrapolate our results to all landowners in the Southern Plains.” (lines 127-129). This sounds a bit like "we couldn't do it, therefore it was not necessary", but the possible response biases (e.g. age) and their likely an influence on the findings needs to be properly acknowledged and dealt with. The statistical analyses used assume that the data represent a random sample from the population, and if this assumption is violated then the inferences drawn from the sample cannot be applied to the population.

I identified two (possible) sources of sampling bias that are likely to have influenced the results of the analyses and these need to be better discussed and taken into account when drawing conclusions from the research.

The first is that the overall sample was chosen as a random subset of 100 landowners in a county, but to this the authors added 318 purposively chosen members of PBAs (making up 16.6 % of the total questionnaires mailed and > 30 % of the responses received). The authors do not tell the reader what percentage the 100 randomly selected landowners per county represents of the total eligible population (landowners with a minimum 100 acre property size). Similarly, it is not clear what percentage of PBA members is represented by the 318 PBA members that were contacted. Including the PBA members as part of the otherwise random sample violates the assumptions of the regression analyses that used to analyse the influence of various demographic factors on woody cover preference and use of fire. The magnitude of the possible bias this introduces is impossible to estimate since no information is given on the overall ratio of PBA members to the general population, and how this compares to the ratio of respondents in the data (> 30 % of the survey respondents were PBA members who were purposively selected as survey participants).

Was there any exploration of what led PBA members to become members of PBAs? One of the study’s findings if that PBA members were more likely to have used fire in the past, but what can one infer about cause and effect? Would landowners given to using fire not be more likely to join a PBA in the first place? Or does joining a PBA result in landowners being more comfortable using fire?

The second possible source of bias is the use of a mail survey. I don’t know what the current situation is in the rural areas of Texas and Oklahoma, but in many parts of the world few people read and respond to letters via “snail” mail. There are likely to be valid reasons for choosing this method, but they need to be justified and explained, and possible biases that may have arisen need to be discussed, ideally with reference to the international literature. I would expect, for example, that older landowners would be more likely to respond to a mail survey. The sample is very strongly skewed towards older respondents but it is unclear whether this reflects the demographics of landowners in the area or whether this is a respondent bias part of the survey. If the latter then the survey results may under-represent the perceptions, preferences and practices of younger farmers who are significantly different in these respects. PBA members made up a third of respondents, what is their age profile compared to that of the non-PBA members? At the very least, the authors should make an effort to compare this to the actual demographics of the study area, which should be available from population census or similar data, to see if a significant age bias exists.

Both these sources of (possible) bias have a bearing on the results reported. Age was a significant predictor of woody cover preference, and both age and PBA membership were found to significantly predict use of prescribed burning. Without consideration (and where possible, elimination) of sampling bias, these results cannot be considered to be robust.

3. Inclusion of woody cover trend data from Hartfield & van Leeuwen (2018)

The obvious gap when trying to determine whether farmer’s perception of current woody cover and past change are accurate is some objective measure of actual cover. The authors address this by using data from a remote sensing study (Hartfield & van Leeuwen 2018) to compare perceptions of change to published rates of change on a per-county basis. While this is a good idea, there are several problems with how this was done and reported.

The first issue concerns the scale mismatch between the perceptions at the individual landowner level and the change data at the level of the county. The authors explain that this was unavoidable, but the implications are not adequately explored. No data of actual woody cover in each county, and some indication of how variable it is in space, are presented. If woody cover in a county varies a lot spatially, then linking it to individual landowner responses will be a bit of a stretch.

The survey was done in 2015, but the study that is cited as the source of the actual change data was published a couple of years later in 2018. I assume that the remote sensing study therefore did not inform the choice of counties for conducting the survey, but that the data were later accessed to compare perceptions of change to actual change. This is a pity, since the motivation for the survey was that woody encroachment is a significant issue, whereas the remote sensing study found that many counties did not actually experience an increase in woody cover. Possible reasons for the discrepancy between landowners’ perceptions are discussed, but in a rather speculative fashion.

I was unable to ascertain where the authors obtained the data shown in Table 7. No source is given in the table caption, but the source is indicated in the text to be Hartfield & van Leeuwen (2018). I looked at the Hartfield & van Leeuwen (2018) paper and the data do not appear in it. Instead, Hartfield and van Leeuwen state in their discussion that “In a future paper, we will use estimates of woody cover from two dates, to compute the change in woody cover for each pixel in the two ecoregions.” So where do the data cited in the submitted manuscript come from?

Since the authors did manage to access woody change cover for the counties in which their survey was done (and will hopefully be able to provide a correct citation), they should also have been able to source and report woody cover in 2004 and 2014, the latter being the year before the survey. This would allow not only comparison of woody cover estimates and actual woody cover, but also whether actual woody cover in a county influenced perceptions and use of prescribed burning.

Lastly, the perceptions of trends and the comparison with published woody cover trends should have been reported in the results section. Instead, they come as something of a surprise in the form of a table and accompanying text at the beginning of the discussion section. The discussion should not introduce new findings but needs to interpret the findings reported in the results section.

Other comments

The introduction reads as if the paper is aimed at a USA-based rangeland science and management audience. For publication in an international journal, the introduction is rather narrowly focused on a local problem. As someone not based in the USA (but interested in woody encroachment and rangeland management), I would have liked to see more international literature to contextualize the situation in Great Plains. The discussion should similarly draw conclusions that go beyond the regional.

Line 32: Were landowners just asked about the “amount” or also about perceived trends?

Lines 36 and 82: “lability” should be “liability”

Line 39: Does membership of PBA cause landowners to be more likely to use fire, or do landowners who are comfortable using fire join PBAs more readily?

Lines 44-46: This final sentence of the abstract is rather vague.

Line 50: The values of woody cover increase given here contrast with the declines in woody cover reported in Table 7.

Lines 55-56: Use the singular (“climate change” and “land use”)?

Lines 59-61: Why were PBA members explicitly included in the survey while membership of FMAs and the other organisations cited was seemingly not considered or explored at all? Also, since FMAs and NRCS are never mentioned again, there is no need to introduce the abbreviations here.

Line 71: The argument about under- or overestimating woody cover is valid but in the absence of actual woody cover data, the study does not actually determine whether farmers under- or over-estimated woody cove on their land.

Lines 84-85: The first question is about landowner perceptions about woody plant expansion and how it correlates with actual trends but much of the focus of the study (and the reported analyses) is farmer’s estimation of current woody cover (rather than trend). “Trend” and “current cover” are not always clearly distinguished in the writing.

Lines 96 and 99: Change “included” to “were”. “Included” implies that the subsequent list is a subset, when in fact all counties sampled are listed.

Lines 116-117: What were the “five areas of enquiry including attitudes concerning woody plants and fire” and are they all reported on in this manuscript? If not, why not?

Line 119 and elsewhere: Journal requirements include the use of SI units. Acres are not SI units. The values given need to be converted to hectares, or at least report equivalent values in SI units for the benefit of readers accustomed to the metric system.

Line 119: What proportion of landowners with properties > 100 acres does the sample of 1600 represent?

Line 120: What is the basis for the 100-acre minimum? Can the choice of this minimum value, and the assumption that owners of smaller properties would be less likely to use fire, somehow be substantiated (e.g. with a reference from the literature)?

Line 123: “Pulled from” seems rather colloquial. “Extracted from”?

Lines 124-125 – what percentage of PBA members do the 126 / 192 members from TX /OK represent?

Line 131: “Participants” needs an apostrophe.

Line 132: Close bracket after “trees”.

Line 143: What this a “yes/no” type of question?

Line 178: Should “assumption” be replaced with “prediction” or “hypothesis”?

Table 3, last row: OK is abbreviated here but this is the only instance – keep use of appreviations consistent or avoid them.

Lines 198-199 – how do the sample demographics compare to the general population? Also, all results should be consistently written in the past tense.

Line 204: But this figure is influenced by the fact that PBA members were actively targeted. What is the rate of PBA membership actually like among landowners?

Table 4. It would be useful to also include what percentage of respondents fall within the "ranching", "heritage", "hunting" and "farmer" land use categories.

Paragraph following Line 207: It would be useful to include the published data on woody cover and trends in each county here and analyse the differences by county.

Line 240: "Reported" is ambiguous – does this refer to woody cover reported by respondents or in the literature? Would “perceived” not be a more appropriate word here than “reported”?

Tables 5 and 6: p-values are not 0.000 – they are greater than zero and should be presented as < 0.001.

Line 248: “low-cost, effective”: Is that how it was put in the questionnaire? If not, I would leave these value-laden terms out of the results section.

Line 250: delete “By contrast”.

Lines 255-256: Rather state this more intuitively, e.g. "older land owners were significantly less likely to have used fire...". The sentence needs a full stop.

Lines 261 onwards: This needs to be reported in the results section, not the discussion.

Table 7 needs to cite the source of the woody cover change data. Also include actual woody cover in 2004 and 2014 for each county, as these values are of interest. For example, a small decline in woody cover might be outweighed in people’s perceptions by overall high woody cover values.

Line 279: Any data / evidence / literature to support this?

Line 283: farmers may be frustrated by the woody cover, but the reader is frustrated by the absence of data on actual woody cover.

Lines 285-287: What does that mean? Not very enlightening when put like this unless one goes and reads the paper (beyond its title). Not only is it unclear what argument the cited paper makes, but there is also no explanation of how/why this would influence perceptions of actual woody cover.

Lines 295 and 297: It would be useful here to know what the percentage of landowners of the rancher, farmer, heritage, hunter types was in in each state.

Line 298: Replace the question mark with a full stop.

Line 322: From what I can ascertain, actual willingness was not measured – “used fire more frequently” would seem more accurate here than “more willing”.

Line 235: Did the questionnaire not probe landowners’ reasons for using / not using fire?

Line 235: What is meant by “legitimate” in this context?

Line 332: How about training and support in the use of fire?

Figure 1: For the benefit of an international audience, please label the states of Texas and Oklahoma.

6. PLOS authors have the option to publish the peer review history of their article (what does this mean?). If published, this will include your full peer review and any attached files.

Reviewer #1: No

Reviewer #2: No

Reviewer #3: No

---

## [Author Response · Author response to Decision Letter 0]

5 Feb 2020

Reviewer #1: 

R1.1: Discuss how potential bias was handled

a. Survey recipients included 100 random landowners from each county in the study area, plus PBA members. It seems that including a specific subset of the sample population would potentially influence findings and make it difficult to draw conclusions, especially since PBA members would already understand the value of prescribed fire and therefore are potentially more aware of the presence of woody plants on their property. How was this potential source of bias accounted for during the analyses?

Response R1.1a: Using PBA membership lists, we intentionally included all PBA members in each study county because if we relied on a randomly drawn PBA membership sample drawn from the general tax records, the chance of selecting an adequately large sample of PBA members for comparison with non-member respondents would have been extremely unlikely. One of the objectives of our study was to understand the effect of PBA membership on perspectives about woody plant encroachment and the use of fire. It is unclear how PBA membership would have enhanced awareness of the presence of woody plants on their property; woody plants are equally visible to both members and non-members. It is indeed likely that PBA members have a better understanding of the use of prescribed fire; confirming this with our data is the basis for arguing that greater landowner access to PBA membership could enhance the broader use of this critically needed rangeland management tool. Previous research has also shown that many PBA members were not always supportive of the use of prescribed fire but rather that joining such a landowner organization enabled them to change their mind about prescribed fire being a risky land management tool and, consequently, to decide to apply fire on their own land (Kreuter et al. 2008). Finally, when we report and discuss our findings, we are careful to discuss only in terms of our respondents in order not to extrapolate any possible response bias to a larger population. 

b. Were any other sources of bias tested for? (e.g. violations of assumptions or outliers)

Response R1.1b: We did not test for other potential sources of response bias. Since our data were derived primarily from specified response scales, outliers are not present. Additionally, as indicated above, to avoid extrapolating potential bias, we refer to respondents only, not landowners in general, in our presentation and discussion of results and not extrapolated to the landowner population from which the sample was drawn.

c. How were unanswered questions handled? (i.e. was there any influence created by non-responses to specific items within questionnaire areas of inquiry)

Response R1.1c: 

We used null values for unanswered questions, resulting in some variation of n values for each variable. We based this on the assumption that data were missing completely at random (MCAR) as we could not determine respondent’s reasons for not answering any particular question. Based on this assumption, we did not impute MCAR data. 

R1.2: Manuscript organization

a. Information in the Results section should only report results; discussion should be located in the Discussion section (line 240-243)

Response R1.2a: All discussion points originally included in the results section were moved to the discussion. 

b. Data should be reported in the Results section (Table 7)

Response R1.2b: Table 7 (now Table 5) and the associated text have been moved for the results section.

R1.3: Results relationship to Discussion and Conclusions

a. It is important to use the Discussion and Conclusion section to show how the study complements the existing literature and answers the research questions. More of this study’s research findings should be listed, as well as referencing tables and figures from the Results sections.

Response R1.23a: We have expanded the discussion section to address additional significant findings (ex: age variable influence on woody cover preference and use of fire) and have connected our findings back to the literature more thoroughly, including more international literature.

R1.4: Data/analyses transparency

a. What software package was used for data analyses?

Response R1.4a: We used STATA 13 and have included that information in the Methodology section and the works cited 

b. Insert table to show questionnaire items that composed attitudes concerning woody plants and fire. This would provide context on the number and type of questions defining each construct and would be an opportunity to indicate response %. Also indicate which items were positively and negatively worded for purpose of reverse-scoring in the data analyses.

Response R1.4b: We have added an abbreviated questionnaire to the supplemental information so that readers can read the questions asked of participants. We only developed latent index variables for ascertaining respondent’s landownership motivations as outlined in Table 2. The question concerning woody cover preference is detailed in Table 1 under the dependent variable label and again in Figure 2 where readers can see the question prompt as it appears in the questionnaire instrument. 

c. Table 2: What type of rotation was used? List rotated factor loading results for all items. Include % of Variance as a separate row.

Response R1.4c: Requested changes have been included in Table 2.

d. Table 5 and 6: Include standard errors for β coeff. Consider reporting odds ratio and confidence interval instead of % Δ odds, as this will help with interpretation of the results.

Response R1.4d: SE of β coeff. Have been included as suggested. We believe that it is simply a matter of interpretation wording preference since % change in odds is simply subtracting the odds ratio from 1 and multiplying the difference by 100. Either method (odds ratio or % change in odds) will provide the same information. Our preferred standard reporting method is % change in odds. 

R1.5: Include supporting references 

Line 284; line 296-7 (in that area – statewide?); line 315; line 241-3 (how does this relate to Table 4?); line 315-6

Response R1.5: Line 284 has been deleted from the manuscript per other reviewer comments. Line 296-7 reference has been added. Line 315 references have been added. Line 241-243 has been removed from the results section and moved to the discussion section and references have been added. Line 315-316 references concerning increasing wildfire risk have been added. 

R1.6: Clarity of statements

a. line 250, 254 – be careful of interpreting % Δ odds as probability; reporting the odds ratio will allow for more accurate interpretation of data

Response R1.6a: Please see response R1.4d above 

b. line 282-3 – what about the possible influence of PBA member bias?

Response R1.6b: Please see R1.1a above.

c. line 286 – expand on this thought

Response R1.6c: We have deleted the sentence and the preceding one as these are perhaps too speculative. 

d. line 303-5 – how does this species relate to the study, or include reference for why it should be specifically mentioned

Response R1.6d: We have added references concerning landowner preference for retaining oak trees on their land (Olenick et al 2005) (Juzwick et al 2011)

e. line 314 – discuss how/why would it escalate wildfire risk

Response R1.6e: We have added language clarifying why cedar/juniper expansion increases risk of wildfire and added 2 references as support. (Brockway et al 2009 and Twidwell et al 2013)

f. line 344 – what preceding questions? 

Response R1.6f: That sentence has been reworded for clarity

R1.7: Completeness of reporting information

Discuss all statistically significant findings – ex. age relationship and woody cover preference (Table 5) – all other significant findings were mentioned, including age for Table 6

Response R1.7: We have added language to the results section about the significant age variable in Table 5 (now Table 6)

R1.8: Proofreading

Several typos and grammatical errors exist and need corrected

Response R1.8: We have carefully proof read the revised manuscript to ensure correct grammar and eliminate typos.

Reviewer #2: 

R2.1: In lines143-148 you note that you do not conduct an assessment of cover change on respondent's properties--a weakness--but you adequately attempt to qualify your results / observations. 

Response R2.1: Since there was no recommendation for change, we did not change the text.

R2.2: L120 has an extra 'to', and presumably L129 is missing a 'not'

Response R2.2: Corrections included.

Reviewer #3: 

R3.1. It seems odd to conduct this research on issues surrounding woody cover increase in counties that actually have had decreasing woody cover (Table 7). Was there any baseline data on woody cover and woody cover change in the study area that informed the study and choice of study sites?

Response R3.1. The spatial analysis of woody plant cover changes in the Southern Plains that concluded some counties experienced woody plant reduction was being conducted at the same time as the landowner survey. These assessments were two parts of a larger research project. Generally, woody plant expansion is thought to have occurred throughout the Southern Plains. We included explanations for the disparity: “The land cover change estimates may have been influenced by the effects of a severe drought between 2010 and 2015 in that dead standing trees, which were more abundant in the Edwards Platea, may have been undetected from space whereas landowners could easily see them from ground level. Additionally, landowners are also able to see small saplings that indicate woody plant increase but these may also not be detectable from space. Conversely, some respondents may have erroneously indicated woody plant expansion because of their growing frustration with the higher than desired woody plant concentration on their land. This may be especially the case on the counties sampled in Texas which have more woody plant cover than those in Oklahoma. Additionally, Roberts et al. (2018) discovered that nearly all policies aimed at containing juniper tend to promote rather than contain the spread of the species due to “doublethink and policy-ecology mismatches.”(17)” 

R3.2: Incomplete information on the questionnaire

a. It would have been useful to have access to the whole (blank) questionnaire. This would allow readers to assess how the questions were phrased, and whether some information that would have been useful to have (e.g. reasons given for using/not using prescribed burning) was collected but not reported, or whether it was not collected. It would also allow for repeatability of the work.

Response R3.2a. We are including an abbreviated copy of the questionnaire in the supplemental information (S1). 

b. Are there substantial aspects of the questionnaire that are not reported in this manuscript? If so, what are the reasons for this? I feel that I am getting an incomplete picture of the questionnaire and its overall aims. Are they the same as reported in this paper or is the work presented here just a subset?

Response R3.2b. Please see the previous response. Not all of the information from a multi- questionnaire aimed at addressing multiple research question can be presented in one journal manuscript. We provide the reference for one other publication so far above and cite it in the revised manuscript. as it includes aspects that have been published elsewhere. See: Kreuter, U.P., D.A. Stroman, C. Wonkka, J. Weir, A.A. Abney, J.K. Hoffman. 2019. Landowner perceptions of legal liability for using prescribed fire in the Southern Plains, USA. Rangeland Ecology & Management. 72: 959-967. This very recent publication has been added in the discussion of liability concerns. 

c. It would have been useful to have asked farmers to estimate the woody cover on their farms at some earlier date using the sample schematics used to elicit perceptions of present and desirable woody cover.

Response R3.2b. We did ask survey participants to respond to the following three statements, although they were not asked to refer to the schematic in selecting their response option, which could have been beneficial: Cedar/juniper has increased on my property in the last ten years; Mesquite has increased on my property in the last ten years; Other woody plants have increased on my property in the last ten years. We report this information in the revised manuscript.

d. The authors state that the survey data cannot be freely shared because of ethics/confidentiality concerns, but this restriction should not apply to the actual questionnaire that was used.

Response R3.2d. Please see Response 3.2a. and 3.2b.

R3.3: Potential sampling bias resulting from the study design

a. The authors simply state that “A non-response bias analysis was not conducted because alternative contact information (e.g., telephone number) required for such an analysis was not obtainable. Due to this limitation, we do extrapolate our results to all landowners in the Southern Plains.” (lines 127-129). This sounds a bit like "we couldn't do it, therefore it was not necessary", but the possible response biases (e.g. age) and their likely an influence on the findings needs to be properly acknowledged and dealt with. The statistical analyses used assume that the data represent a random sample from the population, and if this assumption is violated then the inferences drawn from the sample cannot be applied to the population.

Response R3.3a. See response in R3.3c. We also changed the wording in line 127-129 to read “Due to this limitation, we do not extrapolate our results to all landowners in the Southern Plains.”

b. I identified two (possible) sources of sampling bias that are likely to have influenced the results of the analyses and these need to be better discussed and taken into account when drawing conclusions from the research.

Response R3.3b. See response in R3.3c

c. The first is that the overall sample was chosen as a random subset of 100 landowners in a county, but to this the authors added 318 purposively chosen members of PBAs (making up 16.6 % of the total questionnaires mailed and > 30 % of the responses received). The authors do not tell the reader what percentage the 100 randomly selected landowners per county represents of the total eligible population (landowners with a minimum 100 acre property size). Similarly, it is not clear what percentage of PBA members is represented by the 318 PBA members that were contacted. Including the PBA members as part of the otherwise random sample violates the assumptions of the regression analyses that used to analyze the influence of various demographic factors on woody cover preference and use of fire. The magnitude of the possible bias this introduces is impossible to estimate since no information is given on the overall ratio of PBA members to the general population, and how this compares to the ratio of respondents in the data (> 30 % of the survey respondents were PBA members who were purposively selected as survey participants).

Response R3.3c. 

First, it is impossible to report the total eligible population of landowners in a county who own 100 acres of land or more because many landowners own multiple plats that equal or are greater than 100 acres. We compiled a list of all landowners, according to county tax records, who met the minimum acreage threshold, eliminated any duplicate names and pulled our sample from there. Second, had we relied only on the county tax records to randomly draw landowners who are PBA members, we would have obtained a very small sample of such landowners. Since we were interested in exploring differences in prescribed fire perspectives between PBA member and non-member landowners, we needed to ensure an adequate number of PBA member landowners in our study. Therefore, whereas the non-member landowners represent a random sample of landowners who own at least one plat of land = > 100 acres, the survey participants who were PBA members represent the entire population of PBA members who reside in the counties we surveyed. 

d. Was there any exploration of what led PBA members to become members of PBAs? One of the study’s findings if that PBA members were more likely to have used fire in the past, but what can one infer about cause and effect? Would landowners given to using fire not be more likely to join a PBA in the first place? Or does joining a PBA result in landowners being more comfortable using fire?

Response R3.3d. In this study, we did not inquire about motivations for becoming a PBA member. Previous research indicates that membership motivations vary including safe fire training, access to fire management equipment on burn days, assistance with application of prescribed fire, and access to prescribed fire liability insurance. However, we did ask some questions about how the PBA influences their social networks. While we do not know from this study if joining a PBA resulted in more landowners being more comfortable using fire, previous research (Kreuter et al. 2008, Toledo et al. 2014) as well as this study show there is strong correlation between PBA membership and application of fire on the members own property and assistance with the application of prescribed fire on other people’s properties.

e. The second possible source of bias is the use of a mail survey. I don’t know what the current situation is in the rural areas of Texas and Oklahoma, but in many parts of the world few people read and respond to letters via “snail” mail. There are likely to be valid reasons for choosing this method, but they need to be justified and explained, and possible biases that may have arisen need to be discussed, ideally with reference to the international literature. I would expect, for example, that older landowners would be more likely to respond to a mail survey. The sample is very strongly skewed towards older respondents but it is unclear whether this reflects the demographics of landowners in the area or whether this is a respondent bias part of the survey. If the latter then the survey results may under-represent the perceptions, preferences and practices of younger farmers who are significantly different in these respects. PBA members made up a third of respondents, what is their age profile compared to that of the non-PBA members? At the very least, the authors should make an effort to compare this to the actual demographics of the study area, which should be available from population census or similar data, to see if a significant age bias exists.

Response R3.3e. We have added information about age to the discussion, including several references. The reported average age (66) is in line with several similar studies both within our study area and throughout the U.S. (see references in discussion section). Our age range of respondents was between 30-93, which indicates that the use of a mail survey is still an accessible method for reaching private landowners of all ages. County land ownership databases in our study area do not include information such as phone or email contacts that may allow for other survey methods. In addition, older landowners may be less likely to answer long form surveys by phone or email. Moreover, we have conducted mail surveys for over 20 years with consistently satisfactory response rates ranging from 30-50% whereas the email/web-based surveys that we have conducted have resulted in extremely low response rates of 5% or less. 

f. Both these sources of (possible) bias have a bearing on the results reported. Age was a significant predictor of woody cover preference, and both age and PBA membership were found to significantly predict use of prescribed burning. Without consideration (and where possible, elimination) of sampling bias, these results cannot be considered to be robust.

Response R3.3f. Please see R1.1a above.

R3.4: Inclusion of woody cover trend data from Hartfield & van Leeuwen (2018)

a. The obvious gap when trying to determine whether farmer’s perception of current woody cover and past change are accurate is some objective measure of actual cover. The authors address this by using data from a remote sensing study (Hartfield & van Leeuwen 2018) to compare perceptions of change to published rates of change on a per-county basis. While this is a good idea, there are several problems with how this was done and reported.

Response R3.4a. See response R3.4c

b. The first issue concerns the scale mismatch between the perceptions at the individual landowner level and the change data at the level of the county. The authors explain that this was unavoidable, but the implications are not adequately explored. No data of actual woody cover in each county, and some indication of how variable it is in space, are presented. If woody cover in a county varies a lot spatially, then linking it to individual landowner responses will be a bit of a stretch.

Response R3.4b. See response R3.4c

c. The survey was done in 2015, but the study that is cited as the source of the actual change data was published a couple of years later in 2018. I assume that the remote sensing study therefore did not inform the choice of counties for conducting the survey, but that the data were later accessed to compare perceptions of change to actual change. This is a pity, since the motivation for the survey was that woody encroachment is a significant issue, whereas the remote sensing study found that many counties did not actually experience an increase in woody cover. Possible reasons for the discrepancy between landowners’ perceptions are discussed, but in a rather speculative fashion.

Response R3.4c. We did not have access to the woody cover data cited in the paper during project design. However we have added data from the same source as the % cover change about actual woody plant coverage by county into Table 5. 

d. I was unable to ascertain where the authors obtained the data shown in Table 7. No source is given in the table caption, but the source is indicated in the text to be Hartfield & van Leeuwen (2018). I looked at the Hartfield & van Leeuwen (2018) paper and the data do not appear in it. Instead, Hartfield and van Leeuwen state in their discussion that “In a future paper, we will use estimates of woody cover from two dates, to compute the change in woody cover for each pixel in the two ecoregions.” So where do the data cited in the submitted manuscript come from?

Response R3.4e. The data were obtained directly from Hartfield and van Leeuwen, who were Co-PIs on a larger project of which our survey was one of our components.

e. Since the authors did manage to access woody change cover for the counties in which their survey was done (and will hopefully be able to provide a correct citation), they should also have been able to source and report woody cover in 2004 and 2014, the latter being the year before the survey. This would allow not only comparison of woody cover estimates and actual woody cover, but also whether actual woody cover in a county influenced perceptions and use of prescribed burning.

Response R3.4e. We have added in the woody cover estimates for 2002 and 2014 from the same source as the % change in woody cover. However, we elected not to add county cover estimates into our models concerning perception and use of fire because some respondents may own property in other counties and accordingly their perceptions are not informed only from woody plant cover in the counties in which their sampled property is located. 

f. Lastly, the perceptions of trends and the comparison with published woody cover trends should have been reported in the results section. Instead, they come as something of a surprise in the form of a table and accompanying text at the beginning of the discussion section. The discussion should not introduce new findings but needs to interpret the findings reported in the results section.

Response R3.4a. These results have been moved to the results subsection, “Woody plant cover perceptions and preferences.” 

R3.5: For publication in an international journal, the introduction is rather narrowly focused on a local problem. I would have liked to see more international literature to contextualize the situation in Great Plains. The discussion should similarly draw conclusions that go beyond the regional. 

Response R3.5. We have added more international literature to our discussion in order to draw comparisons with similar issues globally, e.g., Sutherland et al. 2011; Eburn and Cary 2018. 

R3.6: Line 32: Were landowners just asked about the “amount” or also about perceived trends?

Response R3.6. They were also asked about their perceptions about increases in woody plant cover on their land. See R3.24 for more details.

R3.7: Lines 36 and 82: “lability” should be “liability”

Response R3.7. Done

R3.8: Line 39: Does membership of PBA cause landowners to be more likely to use fire, or do landowners who are comfortable using fire join PBAs more readily?

Response R3.8. That is a good question. Because of the requirement by many PBAs for members to assist with burns on other members properties before they receive assistance with a burn on their own property, the first alterative is more likely, i.e., that PBA cause landowners to be more likely to use fire. This is emphasized in the discussion where the following sentence has been added: “it has been found that that PBA’s are very effective in engaging landowners in the use of prescribed fire, because they provide their members with fire safety training and equipment and they promote safe fire application experience among members who assist each other on burn days.” We also added another reference to support this point.

R3.9: Lines 44-46: This final sentence of the abstract is rather vague.

Response R3.9. The sentence has been changed to: “Future outreach efforts to promote prescribed fire to maintain grasslands should more actively support prescribed burn associations, which an effective vehicle for increasing prescribed fire use by private landowners.” 

R3.10: Line 50: The values of woody cover increase given here contrast with the declines in woody cover reported in Table 7.

Response R3.10. To address this issue, we come back to the trends reported by Berg et al. 2015 (Ref #1) in discussing the information derived from Hartfield & van Leeuwen (2018) and presented in Table 5.

R3.11: Lines 55-56: Use the singular (“climate change” and “land use”)?

Response R3.11. Done

R3.12: Lines 59-61: Why were PBA members explicitly included in the survey while membership of FMAs and the other organizations cited was seemingly not considered or explored at all? Also, since FMAs and NRCS are never mentioned again, there is no need to introduce the abbreviations here.

Response R3.12. To avoid confusion, we eliminated the term Fire Management Associations (FMAs), because it is equivalent to Prescribed Burn Associations (PBAs), which is the more generally used term and the term we use throughout the manuscript. We also deleted (NRCS).

R3.13: Line 71: The argument about under- or overestimating woody cover is valid but in the absence of actual woody cover data, the study does not actually determine whether farmers under- or over-estimated woody cove on their land.

Response R3.13. We changed the sentence to: “Accurately estimating woody cover density is difficult from ground level because visual foreshortening of dispersed trees seen from that perspective may suggest higher woody plant density than overhead measurements of canopy cover would indicate.”

R3.14: Lines 84-85: The first question is about landowner perceptions about woody plant expansion and how it correlates with actual trends but much of the focus of the study (and the reported analyses) is farmer’s estimation of current woody cover (rather than trend). “Trend” and “current cover” are not always clearly distinguished in the writing.

Response R3.14. We changed “actual trends” to “estimated changes” which more accurately represents the data we presented in Table 7. We did not use the term “current cover.

R3.15: Lines 96 and 99: Change “included” to “were”. “Included” implies that the subsequent list is a subset, when in fact all counties sampled are listed.

Response R3.15. Done

R3.16: Lines 116-117: What were the “five areas of enquiry including attitudes concerning woody plants and fire” and are they all reported on in this manuscript? If not, why not?

Response R3.16. Given that the not all 5 areas of inquiry are presented in the manuscript, we deleted this sentence as it creates ambiguity.

R3.17: Line 119 and elsewhere: Journal requirements include the use of SI units. Acres are not SI units. The values given need to be converted to hectares, or at least report equivalent values in SI units for the benefit of readers accustomed to the metric system.

Response R3.17. All property sizes were changed to ha with the acres shown in parentheses.

R3.18: Line 119: What proportion of landowners with properties > 100 acres does the sample of 1600 represent?

Response R3.18. 100%. Landowners with less than 100 acres were disqualified from inclusion. 

R3.19: Line 120: What is the basis for the 100-acre minimum? Can the choice of this minimum value, and the assumption that owners of smaller properties would be less likely to use fire, somehow be substantiated (e.g. with a reference from the literature)?

Response R3.19. As we stated in the manuscript: “We applied the minimum acreage requirement with the assumption that landowners with smaller acreage properties are less likely to apply prescribed fire …”. This was based on information gathered during pre-survey meetings with personnel from the Natural Resources Conservation Services, Texas A&M AgriLife Extension, an Oklahoma Extension Service. 

R3.20: Line 123: “Pulled from” seems rather colloquial. “Extracted from”?

Response R3.20. Done.

R3.21: Lines 124-125 – what percentage of PBA members do the 126 / 192 members from TX /OK represent?

Response R3.21. All identified PBA members in each study county were included in the study. If we had relied only on the sample extracted from the county tax records, we would have had a very small number of PBA-member respondents, which would have prevented us from making member/non-member comparisons.

R3.22: Line 131: “Participants” needs an apostrophe.

Response R3.22. Done

R3.23: Line 132: Close bracket after “trees”.

Response R3.23. Done

R3.24: Line 143: Was this a “yes/no” type of question?

Response R3.24. We used the 7-point response scale to answer the question: “To what extent do you DISAGREE OR AGREE with each of the following statements about woody plants?” … with respect to increases in juniper, mesquite and other woody plants during the last 10 years. This is now evident in the survey questionnaire we have added in the Supplemental information. 

R3.25: Line 178: Should “assumption” be replaced with “prediction” or “hypothesis”?

Response R3.25. We replaced “assumption” with “prediction”. 

R3.26: Table 3, last row: OK is abbreviated here but this is the only instance – keep use of abbreviations consistent or avoid them.

Response R3.26. Oklahoma has been spelled out throughout the manuscript.

R3.27: Lines 198-199 – how do the sample demographics compare to the general population? Also, all results should be consistently written in the past tense.

Response R3.27. We are uncertain how we should report general population statistics. We could report general demographic statistics for the 16 study counties or for each of the two states but do not see what this would add to the manuscript. Suffice it to say, that the demographic results provided approximate those from a plethora of other surveys conducted by the researchers during the last 20 years. The past tense has been used throughout in the landowner characteristics results.

R3.28: Line 204: But this figure is influenced by the fact that PBA members were actively targeted. What is the rate of PBA membership actually like among landowners?

Response R3.28. We could not determine which figure in line 204 the reviewer is referring to. No figure was referenced in that line or on that page or the subsequent page. 

R3.29: Table 4. It would be useful to also include what percentage of respondents fall within the "ranching", "heritage", "hunting" and "farmer" land use categories.

Response R3.29. They are not landowner categories they are motivations that landowners have for owning their land. Each respondent may fit into more than one motivation category. We have changed the manuscript to reflect that they are landownership motivations and not categories of landowners (see table 6 and 7). The mean response score for each motivation is also presented in Table 2 to give readers an idea of which motivations were more common. For example, lifestyle/recreation motivations were overall high compared to the farming motivations.

R3.30: Paragraph following Line 207: It would be useful to include the published data on woody cover and trends in each county here and analyze the differences by county.

Response R3.30. Table 7 (now Table 5) and the associated text have been moved to this subsection to address this comment R3.39. 

R3.31: Line 235: Did the questionnaire not probe landowners’ reasons for using / not using fire?

Response R3.31. We did not see how this comment relates to the statement on line 235. 

R3.32: Line 235: What is meant by “legitimate” in this context?

Response R3.32. The line reference provided should have been 325. We changed the “legitimate” to “safe and effective.” 

R3.33: Line 240: "Reported" is ambiguous – does this refer to woody cover reported by respondents or in the literature? Would “perceived” not be a more appropriate word here than “reported”?

Response R3.33. Reported was changed to perceived. 

R3.34: Tables 5 and 6: p-values are not 0.000 – they are greater than zero and should be presented as < 0.001.

Response R3.34. Done

R3.35: Line 248: “low-cost, effective”: Is that how it was put in the questionnaire? If not, I would leave these value-laden terms out of the results section.

Response R3.35. Survey participants were asked the extent to which they agreed with the notion that prescribed fire is “more effective” and “less costly” “than other woody methods for controlling woody plants.” This language was based on the finding by Van Liew et al. 2012, which is cited and refenced in the manuscript. We have added “relatively” to the sentence to connect it to that paper, which is now also cited in this sentence to support the statement. 

R3.36: Line 250: delete “By contrast”.

Response R3.36. Done

R3.37: Lines 255-256: Rather state this more intuitively, e.g. "older land owners were significantly less likely to have used fire...". The sentence needs a full stop.

Response R3.37. Done.

R3.38: Lines 261 onwards: This needs to be reported in the results section, not the discussion.

Response R3.38. Done

R3.39: Table 7 needs to cite the source of the woody cover change data. Also include actual woody cover in 2004 and 2014 for each county, as these values are of interest. For example, a small decline in woody cover might be outweighed in people’s perceptions by overall high woody cover values.

Response R3.39. The source of the woody cover change date was already cited as reference #14 in footnote b. The 2004 and 2014 woody plant cover data have also been added to Table 7 (now Table 5).

R3.40: Line 279: Any data / evidence / literature to support this?

Response R3.40. We have added three references about the effect of drought on woody plant mortality. They are Twidwell et al. 2014, Moore et al. 2016 and Croucher et al. 2019. 

R3.41: Line 283: farmers may be frustrated by the woody cover, but the reader is frustrated by the absence of data on actual woody cover.

Response R3.41. We have deleted the statement. We did provide data about county level woody plant cover (Table 5) as well as landowner perceptions of woody cover on their own land, based on Figure 2. Measuring actual woody cover on each respondent’s land would be extremely difficult and beyond the scope of this study. For this reason, we obtained information about landowner perception of woody plant cover on their land. 

R3.42: Lines 285-287: What does that mean? Not very enlightening when put like this unless one goes and reads the paper (beyond its title). Not only is it unclear what argument the cited paper makes, but there is also no explanation of how/why this would influence perceptions of actual woody cover.

Response R3.42. We have deleted the statement as it relates to the previous sentence.

R3.43: Lines 295 and 297: It would be useful here to know what the percentage of landowners of the rancher, farmer, heritage, hunter types was in in each state.

Response R3.43. These percentages have been included in Table 4.

R3.44: Line 298: Replace the question mark with a full stop.

Response R3.44. Done.

R3.45: Line 322: From what I can ascertain, actual willingness was not measured – “used fire more frequently” would seem more accurate here than “more willing”.

Response R3.45. We changed “more willing” to “appeared to be more accepting” as we did not measure frequency of prescribed fire use either. 

R3.46: Line 332: How about training and support in the use of fire?

Response R3.46. The sentence has been modified to included this suggestion. 

R3.47: Figure 1: For the benefit of an international audience, please label the states of Texas and Oklahoma.

Response R3.47. We have modified Figure 1, including labeling Texas and Oklahoma.

---

## [Decision Letter · Decision Letter 1]

26 Mar 2020

PONE-D-19-20616R1

Landowner perceptions of woody plants and prescribed fire in the southern Great Plains, USA

PLOS ONE

Dear Dr. Stroman,

Thank you for submitting your manuscript to PLOS ONE. After careful consideration, we feel that it has merit but does not fully meet PLOS ONE’s publication criteria as it currently stands. Therefore, we invite you to submit a revised version of the manuscript that addresses the points raised during the review process.

ACADEMIC EDITOR: The issue of non-random participation in the PBA was raised during the first review and it has not been addressed in this revised version.  I agree with this concern clearly emphasized by Reviewer 3.

One statistical tool to compensate for the non-random nature of participation is the use of propensity score matching. Within STATA 13, which the authors are using, this is relatively simple to do these days. Please see: https://www.stata.com/manuals13/teteffectspsmatch.pdf

This step will strengthen the study tremendously. The author should, nonetheless, see this as a suggestion and may take a different analytical approach when addressing the issue of non-random participation.

The authors seem to dismiss the comment raised by the reviewer regarding non-response bias by indicating that because non follow-up phone calls were made this cannot be tested. The authors could conduct a simple comparison between early respondents and later respondents (e.g. first and second waves) and infer non-response bias from differences between these two groups.

See: Armstrong, J.S., Overton, T.S., 1977. Estimating non-response bias in mail surveys. Journal of Marketing Research 14 (3), 396–402.

The authors shall address the minor issues noted by reviewer 1.

We would appreciate receiving your revised manuscript by May 10 2020 11:59PM. To enhance the reproducibility of your results, we recommend that if applicable you deposit your laboratory protocols in protocols.io, where a protocol can be assigned its own identifier (DOI) such that it can be cited independently in the future. For instructions see: http://journals.plos.org/plosone/s/submission-guidelines#loc-laboratory-protocols

We look forward to receiving your revised manuscript.

Kind regards,

Francisco X Aguilar

Academic Editor

PLOS ONE

Reviewers' comments:

Reviewer's Responses to Questions

**Comments to the Author**

1. If the authors have adequately addressed your comments raised in a previous round of review and you feel that this manuscript is now acceptable for publication, you may indicate that here to bypass the “Comments to the Author” section, enter your conflict of interest statement in the “Confidential to Editor” section, and submit your "Accept" recommendation.

Reviewer #1: (No Response)

Reviewer #3: (No Response)

2. Is the manuscript technically sound, and do the data support the conclusions?

Reviewer #1: Partly

Reviewer #3: Partly

3. Has the statistical analysis been performed appropriately and rigorously? 

Reviewer #1: Yes

Reviewer #3: No

4. Have the authors made all data underlying the findings in their manuscript fully available?

Reviewer #1: Yes

Reviewer #3: Yes

5. Is the manuscript presented in an intelligible fashion and written in standard English?

Reviewer #1: Yes

Reviewer #3: Yes

6. Review Comments to the Author

Reviewer #1: R1.1: Discuss how potential bias was handled

a. Survey recipients included 100 random landowners from each county in the study area,

plus PBA members. It seems that including a specific subset of the sample population

would potentially influence findings and make it difficult to draw conclusions, especially

since PBA members would already understand the value of prescribed fire and therefore

are potentially more aware of the presence of woody plants on their property. How was

this potential source of bias accounted for during the analyses?

Response R1.1a: Using PBA membership lists, we intentionally included all PBA

members in each study county because if we relied on a randomly drawn PBA

membership sample drawn from the general tax records, the chance of selecting an

adequately large sample of PBA members for comparison with non-member respondents

would have been extremely unlikely. One of the objectives of our study was to

understand the effect of PBA membership on perspectives about woody plant

encroachment and the use of fire. It is unclear how PBA membership would have

enhanced awareness of the presence of woody plants on their property; woody plants are

equally visible to both members and non-members. It is indeed likely that PBA members

have a better understanding of the use of prescribed fire; confirming this with our data is

the basis for arguing that greater landowner access to PBA membership could enhance

the broader use of this critically needed rangeland management tool. Previous research

has also shown that many PBA members were not always supportive of the use of

prescribed fire but rather that joining such a landowner organization enabled them to

change their mind about prescribed fire being a risky land management tool and,

consequently, to decide to apply fire on their own land (Kreuter et al. 2008). Finally,

when we report and discuss our findings, we are careful to discuss only in terms of our

respondents in order not to extrapolate any possible response bias to a larger population.

- It still appears there is potential sampling bias. When a specific subset of a larger population (PBA members) is sampled and responses combined with a random sampling of the larger population (randomly selected landowners), there is the possibility that the responses from the subset will influence the data set when all responses are analyzed together. It would be more appropriate to analyze each population separately and compare responses from each demographic. If one dataset is used for analyses then a clear description of how this would not influence the results needs to be stated.

b. Were any other sources of bias tested for? (e.g. violations of assumptions or outliers)

Response R1.1b: We did not test for other potential sources of response bias. Since our

data were derived primarily from specified response scales, outliers are not present.

Additionally, as indicated above, to avoid extrapolating potential bias, we refer to

respondents only, not landowners in general, in our presentation and discussion of results

and not extrapolated to the landowner population from which the sample was drawn.

- Providing information on potential bias and how it was handled will help to ensure that results from the data analyses are valid and accurate. Not only does this include outliers (within the data set, not just outside of the specified response scale), but violations of assumptions of statistical tests (e.g. additivity and linearity; normality; homoscedasticity/homogeneity of variance; independence). This information might also help address potential sampling bias.

c. How were unanswered questions handled? (i.e. was there any influence created by nonresponses

to specific items within questionnaire areas of inquiry)

Response R1.1c:

We used null values for unanswered questions, resulting in some variation of n values for

each variable. We based this on the assumption that data were missing completely at

random (MCAR) as we could not determine respondent’s reasons for not answering any

particular question. Based on this assumption, we did not impute MCAR data.

- Again, this is information that should be included in the manuscript.

R1.8: Proofreading

Several typos and grammatical errors exist and need corrected

Response R1.8: We have carefully proof read the revised manuscript to ensure correct

grammar and eliminate typos.

- Grammatical errors and typos persist; continue to revise. Replace colloquial wording with specific language.

Discussion and Conclusion

a. Discussion of perceived v. actual land cover (lines 299-317) – what about potential influence of PBA members due to their higher response rate (i.e. is it possible for PBA members to be more aware of woody encroachment)?

b. Clarify or expand on “doublethink and policy-ecology mismatches” (lines 313-314)

c. Be sure to relate discussion of prescribed fire and woody plant cover back to this research. Lines 354-372 focus on other research; there is an opportunity to relate discussion about PBAs (lines 365-366) to this research (Table 7)

d. Lines 375-376 – include reference

e. Conclusion paragraph (lines 381-392) – be sure to clearly summarize relationships to the three research questions and how this relates to opportunities for future research opportunities

Reviewer #3: The authors have not adequately addressed the concern raised in my review (which was also raised by Reviewer 1) that the addition of purposively sampled members of PBAs to a random sample of landowners results in a sample that, when analysed as a single data set, is not random. To be a truly random sample, every subject in the target population must have an equal chance of being selected in the sample. In the case of the combined random + PBA member data set, this is clearly not the case. I understand the reason for purposively sampling PBA members, as there are relatively few of them, but to include them in the overall analysis violates this key assumption. This is not acknowledged in the revised manuscript and no attempt was made to correct for this in the analyses.

The other possible sources of bias were evidently difficult to eliminate (mail survey – possible age bias), but the authors make a reasonable case in their rebuttal. However, this information which is of interest to the reader is not included in the revised text. All they have done is to state “Due to this limitation, we do NOT extrapolate our results to all landowners in the Southern Plains.” This may well be honest but if the aim was not to extrapolate to the bigger population, why conduct a massive survey?

I understand that the data collection can no longer be changed, and that the data generated from the survey are still valuable and deserve to be published. Nevertheless, in analyzing the data, and reporting and interpreting the results, care needs to be taken to acknowledge and deal with the issues.

For example, I commented:

R3.28: Line 204: But this figure is influenced by the fact that PBA members were actively targeted. What is the rate of PBA membership actually like among landowners?

The “figure” referred to was the value of 32 % in the sentence “About a third of respondents (32%) reported being a member of a PBA.” The percentage of PBA members relative to the general population is inflated by the inclusion of a non-random proportion of PBA members. So what then is the use of knowing that 32 % of the sample are self-reported PBA members? It does not reflect the true percentage of PBA members among the general population. Can this value be used to interpret findings that PBA membership significantly influences attitudes to fire management?

I think the authors need to make a better effort (perhaps with the support of a statistician or data analyst) to address this concern, so that the way the data are reported and interpreted is robust and can allow the reader to draw conclusions about land owners in the greater region, with the appropriate caveats if required. I am not a statistician and while I can see the problem, I cannot offer an alternative way of analyzing the data that gets around the assumption of a random sample – but a statistician or data analyst would no doubt be able to help.

Apart from this I am satisfied with the revisions and author responses.

7. PLOS authors have the option to publish the peer review history of their article (what does this mean?). If published, this will include your full peer review and any attached files.

Reviewer #1: No

Reviewer #3: No

---

## [Author Response · Author response to Decision Letter 1]

20 Jul 2020

PONE-D-19-20616R1 – Response to Reviewer Comments

Comment AE 1.1: The issue of non-random participation in the PBA was raised during the first review and it has not been addressed in this revised version. I agree with this concern clearly emphasized by Reviewer 3. One statistical tool to compensate for the non-random nature of participation is the use of propensity score matching. Within STATA 13, which the authors are using, this is relatively simple to do these days. Please see:https://www.stata.com/manuals13/teteffectspsmatch.pdf. This step will strengthen the study tremendously. The author should, nonetheless, see this as a suggestion and may take a different analytical approach when addressing the issue of non-random participation.

Response AE 1.1: To address the ongoing concern about non-random participation of PBA members, we have rewritten the description of our sampling approach as follows: “The survey included 1,918 landowners in Texas and Oklahoma who owned a minimum of 40.5 ha (100 acres). We applied this minimum size requirement with the assumption that landowners with smaller sized properties are less likely to apply prescribed fire, regardless of their perspectives of woody plant density. This survey sample was derived using a stratified sample selection approach. The first stratum consisted of general landowners included in the tax databases of each of the 16 selected counties. From each of these databases, 100 landowners who owned at least 40.5 ha were randomly selected for a first stratum sub-sample of 1,600 landowners. The second stratum consisted all members of PBA in the 16 selected counties. The population of PBA members in each of the two states was small enough that our PBA sample consisted the whole subgroup. This resulted second stratum sub-sample of 318 PBA members (126 in Texas and 192 in Oklahoma). To ensure that the PBA members were not a biased subset of the combined sample of 1,918 landowners, we compared responses from the two subsamples regarding perceptions of woody plants and prescribed fire and about the proportion of income derived from the property.” 

The results of this analysis are included at the beginning of the results section. We found no significant difference between PBA members and non-members regarding the proportion of annual income derived from the property or about existing and preferred woody plant density (see Table 4 in the manuscript). However, PBA members were more likely to respond that woody plants are a problem on their land (non -members also agreed with that statement but not as strongly), and they were also much more likely to respond that they were in favor of applying prescribed fire to their own land. In the discussion we included the following statement to address these results: “While we did not find any difference between PBA members and non-members in their perceptions and preferences of woody plant densities on their land, we did find that PBA members were more likely to view woody plant expansion as a problem on their property and to apply prescribed fire to address this concern. However, the directionality of this difference is unclear; i.e., whether PBA membership elevated the perception of woody plants as a problem or if landowners who viewed woody plants as a problem were more likely to become a PBA member in order to apply fire using PBA membership benefits including access to labor, fire management equipment and fire training.” 

Comment AE 1.2: The authors seem to dismiss the comment raised by the reviewer regarding non-response bias by indicating that because non follow-up phone calls were made this cannot be tested. The authors could conduct a simple comparison between early respondents and later respondents (e.g. first and second waves) and infer non-response bias from differences between these two groups. See: Armstrong, J.S., Overton, T.S., 1977. Estimating non-response bias in mail surveys. Journal of Marketing Research 14 (3), 396–402.

Response AE 1.2: As previously stated, we were not able to follow up with non-respondents so conducting traditional non-response tests was not possible for this study. Per the reviewers’ recommendation based on Armstrong and Overton (1977), which has been cited over 15,000, we have now attempted to test for potential non-response bias by comparing responses from early and late respondents, even though we could find no evidence that this method has been quantitatively verified to support it as an accurate estimate of non-response bias. Respondents were included in one of two groups based on receipt of their questionnaire during or after the first 3 weeks after the mailing date of the questionnaire (i.e., first and second wave responders). We used Mann-Whitney rank tests to compared the variable values of the two groups for five key variables: age of respondent; reliance on income from their property; perception of and preference for woody plant cover on their land, and whether or not they thought woody plants were a problem on their land. We found no statistically significant differences for any of the five variables between the two groups (p = 0.0707 - 0.7721). However, we are not convinced that testing early versus later respondents is a legitimate method for dealing with non-response bias as all the data are from respondents. Therefore, we have continued to be cautious about extrapolating our results to the broader population of landowners in the 16 counties from which the sample of general landowners was drawn. We have included this concern in the limitations paragraph in the discussion section. 

Comment AE 1.3: The authors shall address the minor issues noted by reviewer 1.

Response AE 1.2: These responses are provided below.

Comment R 1.1: It still appears there is potential sampling bias. When a specific subset of a larger population (PBA members) is sampled and responses combined with a random sampling of the larger population (randomly selected landowners), there is the possibility that the responses from the subset will influence the data set when all responses are analyzed together. It would be more appropriate to analyze each population separately and compare responses from each demographic. If one dataset is used for analyses then a clear description of how this would not influence the results needs to be stated.

Response R 1.1: See response AE 1.1. above.

Previous Comment R1.1: Discuss how potential bias was handled

a. Survey recipients included 100 random landowners from each county in the study area, plus PBA members. It seems that including a specific subset of the sample population would potentially influence findings and make it difficult to draw conclusions, especially since PBA members would already understand the value of prescribed fire and therefore are potentially more aware of the presence of woody plants on their property. How was this potential source of bias accounted for during the analyses?

Previous Response R1.1a: Using PBA membership lists, we intentionally included all PBA members in each study county because if we relied on a randomly drawn PBA membership sample drawn from the general tax records, the chance of selecting an adequately large sample of PBA members for comparison with non-member respondents would have been extremely unlikely. One of the objectives of our study was to understand the effect of PBA membership on perspectives about woody plant encroachment and the use of fire. It is unclear how PBA membership would have enhanced awareness of the presence of woody plants on their property; woody plants are equally visible to both members and non-members. It is indeed likely that PBA members have a better understanding of the use of prescribed fire; confirming this with our data is the basis for arguing that greater landowner access to PBA membership could enhance the broader use of this critically needed rangeland management tool. Previous research has also shown that many PBA members were not always supportive of the use of prescribed fire but rather that joining such a landowner organization enabled them to change their mind about prescribed fire being a risky land management tool and, consequently, to decide to apply fire on their own land (Kreuter et al. 2008). Finally, when we report and discuss our findings, we are careful to discuss only in terms of our respondents in order not to extrapolate any possible response bias to a larger population.

Comment R 1.2: Providing information on potential bias and how it was handled will help to ensure that results from the data analyses are valid and accurate. Not only does this include outliers (within the data set, not just outside of the specified response scale), but violations of assumptions of statistical tests (e.g. additivity and linearity; normality; homoscedasticity/ homogeneity of variance; independence). This information might also help address potential sampling bias.

Response R 1.2: See response AE 1.1 and AE 1.2 above. We do not understand what the reviewer means by outliers within the data set – we had previously stated that the response data were primarily based on specified response options (generally 5).

Previous Comment R1.1b: Were any other sources of bias tested for? (e.g. violations of assumptions or outliers)

Previous Response R1.1b: We did not test for other potential sources of response bias. Since our

data were derived primarily from specified response scales, outliers are not present.

Additionally, as indicated above, to avoid extrapolating potential bias, we refer to

respondents only, not landowners in general, in our presentation and discussion of results

and not extrapolated to the broader landowner population from which the sample was drawn.

Comment R 1.3: Again, this is information that should be included in the manuscript.

Response R 1.3: We have included this information in the methods section of the manuscript as suggested. 

Previous comment R 1.1c. How were unanswered questions handled? (i.e. was there any influence created by nonresponses to specific items within questionnaire areas of inquiry)

Previous Response R1.1c: We used null values for unanswered questions, resulting in some variation of n values for each variable. We based this on the assumption that data were missing completely at random (MCAR) as we could not determine respondent’s reasons for not answering any particular question. Based on this assumption, we did not impute MCAR data.

Comment R 1.4: Grammatical errors and typos persist; continue to revise. Replace colloquial wording with specific language.

Response R 1.4: We have again proofread the entire manuscript and asked the additional co-author, who is a statistician, to also read the document in order to identify the unspecified persistent “grammatical errors” and “colloquialisms” and made changes where appropriate. Without explicit specification, we do not know what other errors and colloquialisms the reviewer might be referring to.

Comment R 1.5: Discussion of perceived v. actual land cover (lines 299-317) – what about potential influence of PBA members due to their higher response rate (i.e. is it possible for PBA members to be more aware of woody encroachment)?

Response R1.5: See response AE 1.1. above.

Comment R 1.6: Clarify or expand on “doublethink and policy-ecology mismatches” (lines 313-314)

Response R1.6: This is a quotation from a cited reference which stated: “Mismatches between policy and ecology can lead natural resource agencies toward a “doublethink” mentality (where contradictory thoughts exist without acknowledged cognitive dissonance) that produces policies that simultaneously promote and control invasive species.” We have replaced the above quote with an expanded explanation based on this statement.

 The new wording reads “Additionally, Roberts et al. (2018) discovered that nearly all policies aimed at containing juniper tend to promote rather than contain the spread of the species due to mismatches between policy and ecology and contradictory programs within natural resources agencies that simultaneously promote and control invasive species” 

Comment R 1.7: Be sure to relate discussion of prescribed fire and woody plant cover back to this research. Lines 354-372 focus on other research; there is an opportunity to relate discussion about PBAs (lines 365-366) to this research (Table 7)

Response R1.7: The line numbers to which the reviewer refers are not included in the revised version of the manuscript that goes only to line 368.

Comment R 1.8: Lines 375-376 – include reference

Response R1.8: Unclear to what the reviewer is referring – in our manuscript the maximum line number is 368.

Comment R1.9: Conclusion paragraph (lines 381-392) – be sure to clearly summarize relationships to the three research questions and how this relates to opportunities for future research opportunities.

Response R1.9: The concluding paragraph has been revised to summarize the relationship to the three research questions and to indicate opportunities for future research. 

Reviewer Comment 3.1: The authors have not adequately addressed the concern raised in my review (which was also raised by Reviewer 1) that the addition of purposively sampled members of PBAs to a random sample of landowners results in a sample that, when analyzed as a single data set, is not random. To be a truly random sample, every subject in the target population must have an equal chance of being selected in the sample. In the case of the combined random + PBA member data set, this is clearly not the case. I understand the reason for purposively sampling PBA members, as there are relatively few of them, but to include them in the overall analysis violates this key assumption. This is not acknowledged in the revised manuscript and no attempt was made to correct for this in the analyses.

Response R3.1: See response AE 1.1. above.

Reviewer Comment 3.2: The other possible sources of bias were evidently difficult to eliminate (mail survey – possible age bias), but the authors make a reasonable case in their rebuttal. However, this information which is of interest to the reader is not included in the revised text. All they have done is to state “Due to this limitation, we do NOT extrapolate our results to all landowners in the Southern Plains.” This may well be honest but if the aim was not to extrapolate to the bigger population, why conduct a massive survey?

Response R3.2: See response AE 1.2 above.

Reviewer Comment 3.3: I understand that the data collection can no longer be changed, and that the data generated from the survey are still valuable and deserve to be published. Nevertheless, in analyzing the data, and reporting and interpreting the results, care needs to be taken to acknowledge and deal with the issues. For example, I commented: R3.28: Line 204: But this figure is influenced by the fact that PBA members were actively targeted. What is the rate of PBA membership actually like among landowners? The “figure” referred to was the value of 32 % in the sentence “About a third of respondents (32%) reported being a member of a PBA.” The percentage of PBA members relative to the general population is inflated by the inclusion of a non-random proportion of PBA members. So what then is the use of knowing that 32 % of the sample are self-reported PBA members? It does not reflect the true percentage of PBA members among the general population. Can this value be used to interpret findings that PBA membership significantly influences attitudes to fire management? I think the authors need to make a better effort (perhaps with the support of a statistician or data analyst) to address this concern, so that the way the data are reported and interpreted is robust and can allow the reader to draw conclusions about land owners in the greater region, with the appropriate caveats if required. I am not a statistician and while I can see the problem, I cannot offer an alternative way of analyzing the data that gets around the assumption of a random sample – but a statistician or data analyst would no doubt be able to help.

Response R3.3: We have consulted with a statistician, who is now a co-author of the manuscript, about the potential bias concerns. The statistician helped us craft revisions in the manuscript and the preceding responses to address this issue to the extent possible. We have also addressed these issues in a limitations and future research paragraph at the end of the discussion section.

---

## [Editor Report · Decision Letter 2]

5 Aug 2020

PONE-D-19-20616R2

Landowner perceptions of woody plants and prescribed fire in the Southern Plains, USA

PLOS ONE

Dear Dr. Stroman,

Thank you for submitting your manuscript to PLOS ONE. After careful consideration, we feel that it has merit but does not fully meet PLOS ONE’s publication criteria as it currently stands. Therefore, we invite you to submit a revised version of the manuscript that addresses the points raised during the review process.

We look forward to receiving your revised manuscript.

Kind regards,

Francisco X Aguilar

Academic Editor

PLOS ONE

Additional Editor Comments:

The authors have engaged in a major revision of the study. Although some differences remain between the reviewers' identified criticisms and the authors' approach to dealing with them, I deemed these to be acceptable.

As a minor revision, I will suggest the authors have a separate 'Conclusions', from a 'Discussion and Conclusion', section. This would largely be the last paragraph in the current section.

As the authors engage in this minor revision, I will also ask them to try to offer clear answers in the Conclusions section to their three specific questions: 1) Do landowner perceptions about woody plant expansion correspond with estimated changes in their area? 2) Do landowners within the Southern Plains desire less woody cover on their land? 3) Do expressed woody cover preferences influence the adoption of prescribed fire on private land?

This might seem like a small change but it will help with readability and the potential impact of this paper.

We appreciate you considering PLOS One as an outlet for your research.

---

## [Author Response · Author response to Decision Letter 2]

20 Aug 2020

Editor comment:

The authors have engaged in a major revision of the study. Although some differences remain between the reviewers' identified criticisms and the authors' approach to dealing with them, I deemed these to be acceptable.

As a minor revision, I will suggest the authors have a separate 'Conclusions', from a 'Discussion and Conclusion', section. This would largely be the last paragraph in the current section.

As the authors engage in this minor revision, I will also ask them to try to offer clear answers in the Conclusions section to their three specific questions: 1) Do landowner perceptions about woody plant expansion correspond with estimated changes in their area? 2) Do landowners within the Southern Plains desire less woody cover on their land? 3) Do expressed woody cover preferences influence the adoption of prescribed fire on private land?

This might seem like a small change but it will help with readability and the potential impact of this paper.

Response: We have made the requested changes in the manuscript, focusing the conclusion section on the 3 research questions.

---

## [Editor Report · Decision Letter 3]

24 Aug 2020

Landowner perceptions of woody plants and prescribed fire in the Southern Plains, USA

PONE-D-19-20616R3

Dear Dr. Stroman,

We’re pleased to inform you that your manuscript has been judged scientifically suitable for publication and will be formally accepted for publication once it meets all outstanding technical requirements.

Kind regards,

Francisco X Aguilar

Academic Editor

PLOS ONE

---

## [Editor Report · Acceptance letter]

27 Aug 2020

PONE-D-19-20616R3 

Landowner perceptions of woody plants and prescribed fire in the Southern Plains, USA 

Dear Dr. Stroman:

I'm pleased to inform you that your manuscript has been deemed suitable for publication in PLOS ONE. Congratulations! Your manuscript is now with our production department. 

Kind regards, 

on behalf of

Dr. Francisco X Aguilar 

Academic Editor

PLOS ONE